# Accuracy and data efficiency in deep learning models of protein expression

Evangelos-Marios Nikolados[1], Arin Wongprommoon[1], Oisin Mac Aodha[2,3], Guillaume Cambray[4,5] & Diego A. Oyarzún [1,2,3] ✉

Synthetic biology often involves engineering microbial strains to express high-value proteins. Thanks to progress in rapid DNA synthesis and sequencing, deep learning has emerged as a promising approach to build sequence-to-expression models for strain optimization. But such models need large and costly training data that create steep entry barriers for many laboratories. Here we study the relation between accuracy and data efficiency in an atlas of machine learning models trained on datasets of varied size and sequence diversity. We show that deep learning can achieve good prediction accuracy with much smaller datasets than previously thought. We demonstrate that controlled sequence diversity leads to substantial gains in data efficiency and employed Explainable AI to show that convolutional neural networks can finely discriminate between input DNA sequences. Our results provide guidelines for designing genotype-phenotype screens that balance cost and quality of training data, thus helping promote the wider adoption of deep learning in the biotechnology sector.

Microbial production systems have found applications in many sectors of the economy[1]. In a typical microbial engineering pipeline, cellular hosts are transformed with heterologous genes that code for target protein products, and a key requirement is maximization of titers, productivity, and yield. Such optimization requires the design of genetic elements that ensure high transcriptional and translational efficiency[2], such as promoter[3] or ribosomal binding sequences[4]. However, prediction of protein expression is notoriously challenging and, as a result, strain development suffers from costly rounds of prototyping and characterization, typically relying on heuristic rules to navigate the sequence space towards increased production.

Progress in batch DNA synthesis and high-throughput sequencing has fueled the use of deep mutational scanning to study genotype-phenotype associations. Several works have combined high-throughput mutagenesis with a diverse range of measurable phenotypes, including protein expression[5–8], ribosome loading[9], and DNA methylation[10,11]. As a result, recent years have witnessed a substantial interest in machine learning methods that leverage such data for phenotypic prediction[9,12–15]. In synthetic biology, recent works have incorporated machine learning into the design-build-test cycle for predictive modelling of ribosomal binding sequences[16], RNA constructs[17], promoters[18] and other regulatory elements[19]. Such sequence-to-expression models can be employed as in silico platforms for discovering variants with improved expression properties, paving the way toward a new level of computer-aided design of production strains[18].

Deep learning algorithms, in particular, can uncover relations in the data on a scale that would be impossible by inspection alone, owing to their ability to capture complex dependencies with minimal prior assumptions[20]. Although deep learning models can produce highly accurate phenotypic predictions[12,21,22], they come at the cost of enormous data requirements for training, typically ranging from tens to hundreds of thousands of sequences; see recent examples in Supplementary Table S1. Little attention has been paid to deep learning

[1]School of Biological Sciences, University of Edinburgh, Edinburgh EH9 3JH, UK. [2]School of Informatics, University of Edinburgh, Edinburgh EH8 9AB, UK. [3]The Alan Turing Institute, London NW1 2DB, UK. [4]Diversité des Génomes et Interactions Microorganismes Insectes, University of Montpellier, INRAE UMR 1333, Montpellier, France. [5]Centre de Biologie Structurale, University of Montpellier, INSERM U1054, CNRS UMR5048, Montpellier, France. ✉e-mail: d.oyarzun@ed.ac.uk

models in synthetic biology applications where data sizes are far below the requirements of state-of-the-art algorithms and, moreover, there is a poor grasp of what makes a good training dataset for model training. This is particularly relevant in applications where the cost of strain phenotyping is a limiting factor, as this places an upper ceiling on the number of variants that can be screened. The challenge is then to design a limited set of variants so that the resulting data can be employed to train useful predictors of protein expression. For example, if the sequence space has a broad and shallow coverage, i.e. composed of distant and isolated variants, the resulting data may be difficult to regress because each sample contains little information that correlates with expression. Conversely, if the coverage of the screen is narrow and deep, i.e. composed of closely related sequence variants, models may be accurate but generalize poorly to other regions of the sequence space.

Here, we trained a large number of sequence-to-expression models on datasets of variable size and sequence diversity. We employed a large screen of superfolder GFP-producing (sfGFP) strains in *Escherichia coli*[23] that was designed to ensure a balanced coverage of the sequence space. We sampled these data so as to construct training datasets of varying size and controlled sequence diversity. We first establish the baseline performance of a range of classic, non-deep, machine learning models trained on small datasets with various phenotype distributions and a range of strategies for encoding DNA sequences. This analysis revealed that for this particular dataset, accurate models can be trained on as few as a couple of thousand variants. We moreover show that convolutional neural networks (CNN), a common deep learning architecture, additional improve predictions without the need to acquire additional data. Using tools from Explainable AI[24], we show that CNNs can better discriminate

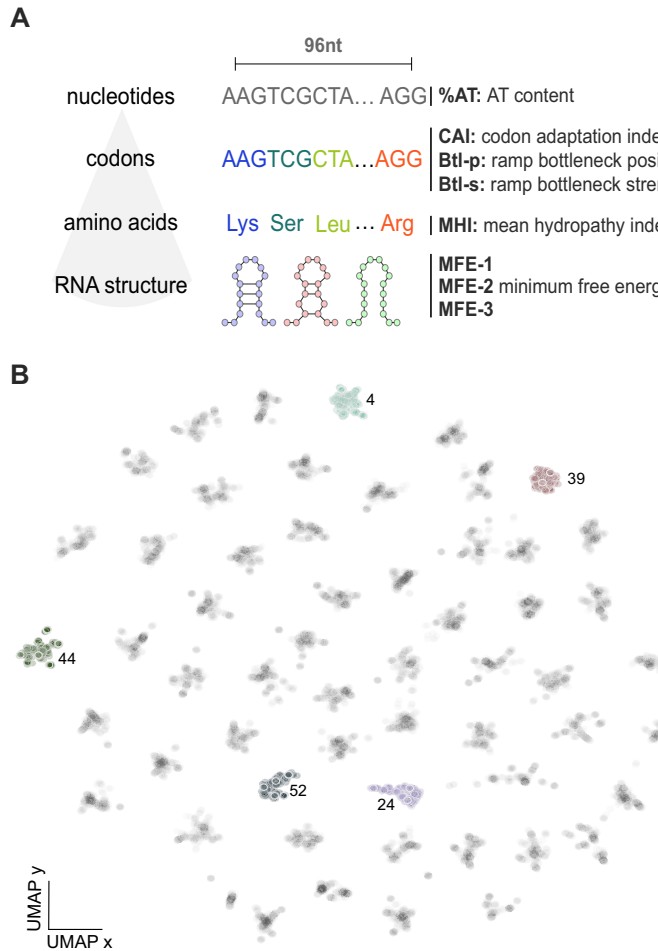

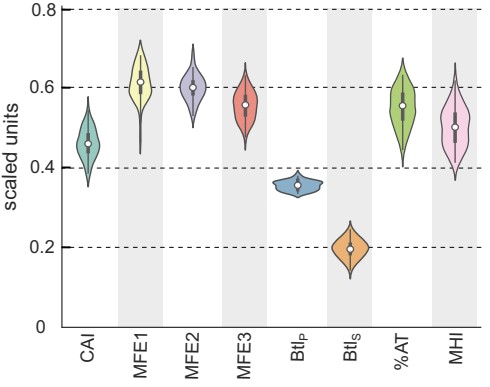

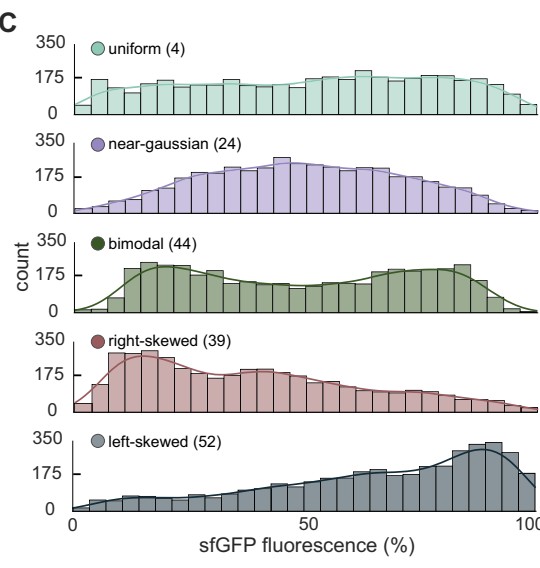

**Fig. 1 | Characterization of the training data. A** We employed a large phenotypic screen in *Escherichia coli*[23] of an sfGFP coding gene preceded by a variable 96nt sequence. The variable region was designed on the basis of eight sequence properties previously described as impacting translational efficiency: nucleotide content (%AT), patterns of codon usage (codon adaptation index, CAI, codon ramp bottleneck position, Btl$_P$, and strength, Btl$_S$), hydrophobicity of the polypeptide (mean hydrophobicity index, MHI) and stability of three secondary structures tiled along the transcript (MFE-1, MFE-2, and MFE-3). A total of 56 seed sequences were designed to provide a broad coverage of the sequence space, and then subjected to controlled randomization to create 56 mutational series of ~4000 sequences each. After removal of variants with missing measurements, the dataset contains 228,000 sequences. Violin plots show the distribution of the average value of the eight properties across the 56 mutational series; the biophysical properties were normalized to the range [0, 1] and then averaged across series. For all violins, the white circle indicates the median, box edges are at the 25th and 75th percentiles, and whiskers show the 95% confidence interval. **B** Two dimensional UMAP[27] visualization of overlapping 4-mers computed for all 228,000 sequences; this representation reveals 56 clusters, with each cluster corresponding to a mutational series that locally explores the sequence space around its seed; we have highlighted five series with markedly distinct phenotype distributions (labels denote the series number). Other UMAP projections for overlapping 3-mers and and 5-mers are shown in Supplementary Fig. S1. **C** Mutational series with qualitatively distinct phenotypic distributions, as measured by FACS-sequencing of sfGFP fluorescence normalized to its maximal measured value; solid lines are Gaussian kernel density estimates of the fluorescence distribution. Measurements are normalized to the maximum sfGFP fluorescence across cells transformed with the same construct averaged over 4 experimental replicates of the whole library[23]. Fluorescence distributions for all mutational series are shown in Supplementary Fig. S2.

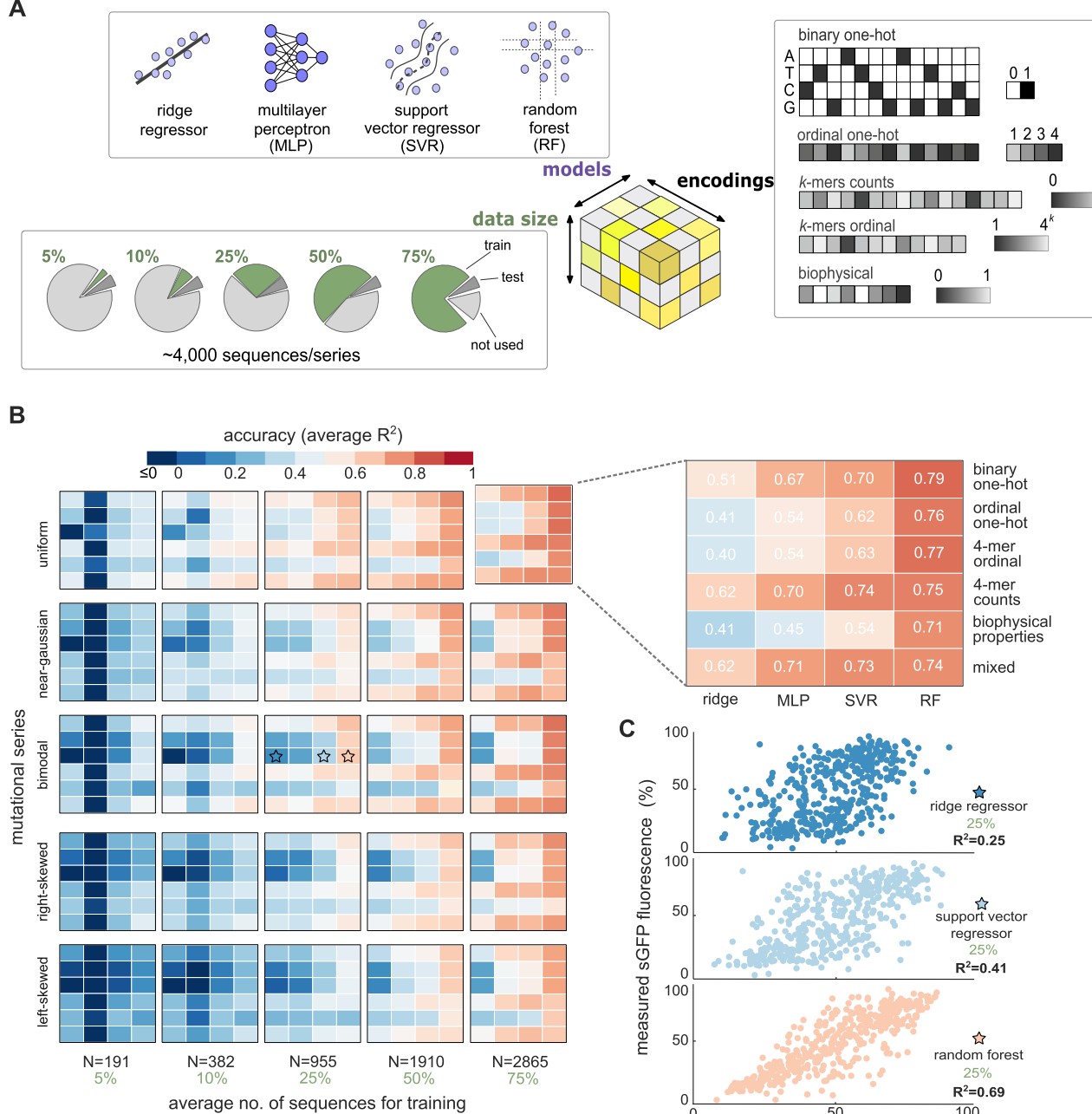

**Fig. 2 | Accuracy of non-deep machine learning models. A** We trained models using datasets of variable size and with different strategies for DNA encoding. Sequences were converted to numerical vectors with five DNA encoding strategies (Table 1), plus an additional mixed encoding consisting of binary one-hot augmented with the biophysical properties of Fig. 1A; in all cases, one-hot encoded matrices were flattened as vectors of dimension 384. We considered four non-deep models trained on an increasing number of sequences from five mutational series with different phenotype distributions (Fig. 1B). **B** Impact of DNA encoding and data size on model accuracy. Overall we found that random forest regressors and binary one-hot encodings provide the best accuracy; we validated this optimal choice across the whole sequence space by training more than 5000 models in all mutational series (Supplementary Fig. S5). Phenotype distributions have a minor impact on model accuracy thanks to the use of stratified sampling for training.

Model accuracy was quantified by the coefficient of determination ($R^2$) between predicted and measured sfGFP fluorescence, computed on ~400 test sequences held-out from training and validation. The reported $R^2$ values are averages across five training repeats with resampled training and test sets (Monte Carlo cross-validation). In each training repeat, we employed the same test set for all models and encodings. The full cross-validation results (Supplementary Fig. S4) show robust performance and little overfitting, particularly for the best performing models. **C** Exemplar predictions on held-out sequences for three models from panel B (marked with stars); the shown models were trained on 25% of mutational series 44 (bimodal fluorescence distribution; Fig. 1C) using 4-mer ordinal encoding. Details on model training and hyperparameter optimization can be found in the Methods, Supplementary Fig. S3, and Supplementary Tables S2–S3.

between input sequences than their non-deep counterparts and, moreover, the convolutional layers provide a mechanism to extract sequence features that are highly predictive of protein expression. We finally demonstrate that in limited data scenarios, controlled sequence diversity can improve data efficiency and predictive performance across larger regions of the sequence space. We validated this conclusion in a recent dataset of ~3000 promoter sequences in *Saccharomyces cerevisiae*[25]. Our results provide a systematic characterization

**Table 1 | DNA encodings for model training**

| DNA encoding | Resolution | Dimension | Positional |
|---|---|---|---|
| Biophysical properties | Global | 8 | × |
| $k$-mer counts | Subsequence | $4^k$ | × |
| $k$-mer ordinal | Subsequence | $L - k + 1$ | ✓ |
| Binary one-hot | Single base | $4L$ | ✓ |
| Ordinal one-hot | Single base | $L$ | ✓ |
| Mixed | Multiple | $4L + 8$ | ✓ |

We considered sequence encodings at three resolutions. In the global encoding, sequences are described by the eight biophysical features employed in the original experimental design[23] (Fig. 1A). At a subsequence resolution, we considered two versions of overlapping $k$-mers: an ordinal version where each k-mer is assigned a unique integer value between 1 and $4 \times k$, and $k$-mer counts containing the number of occurrences of each unique $k$-mer along the sequence; in our results we generally employed $k = 4$ for model training, but observed similar results for other choices of $k$. For base-resolution encodings, we employed two variants of one-hot encoding: binary one-hot where a sequence of length L is encoded as a binary matrix of size $4 \times L$, with each column having a one at the position corresponding to the base in the sequence, and zeros elsewhere; ordinal one-hot encoding assigns a unique integer value to each of the four bases, resulting in encoded vectors of length $L$. Mixed encodings were constructed from flattened one-hot encoded matrices concatenated with the vector of biophysical properties, leading to feature vectors of dimension $4 \times L + 8$.

of sequence-to-expression machine learning models, with implications for the wider adoption of deep learning in strain design and optimization.

## RESULTS

### Size and diversity of training data

We sought to compare various machine learning models using datasets of different size and diversity. To this end, we employed the genotype-phenotype association data from Cambray et al.[23]. This dataset comprises fluorescence measurements for an sfGFP-coding sequence in *Escherichia coli*, fused with more than 240,000 upstream 96nt regions that were designed to perturb translational efficiency and the resulting expression level. The library of upstream sequences was randomized with a rigorous design-of-experiments approach so as to achieve a balanced coverage of the sequence space and a controlled diversity of variants. Specifically, 96nt sequences were designed from 56 seeds with maximal pairwise Hamming distances. Each seed was subject to controlled randomization using the D-Tailor framework[26], so as to produce mutational series with controlled coverage of eight biophysical properties at various levels of granularity: nucleotide sequence, codon sequence, amino acid sequence, and secondary mRNA structure (Fig. 1A).

The complete dataset contains 56 mutational series that provide wide coverage of the sequence space, while each series contains ~4000 sequences for local exploration in the vicinity of the seed. The dataset is particularly well suited for our study because it provides access to controllable sequence diversity, as opposed to screens that consider fully random sequences with limited coverage or single mutational series that lack diversity.

To further characterize sequence diversity across the library of 56 mutational series, we visualized the distribution of overlapping 4-mers using the Uniform Manifold Approximation and Projection (UMAP) algorithm for dimensionality reduction[27]. The resulting two-dimensional distribution of sequences (Fig. 1B) shows a clear structure of 56 clusters, each corresponding to a mutational series. Moreover, the sfGFP fluorescence data (Fig. 1C, Supplementary Fig. S2) display marked qualitative differences across mutational series, including near-Gaussian distributions, left- and right-skewed distributions, as well as bimodal and uniform distributions. This indicates that the dataset is diverse in both genotype and phenotype space, and thus well suited for benchmarking machine learning models because it

allows probing the impact of both genetic and phenotypic variation on model accuracy.

### Impact of DNA encoding and size of training data

To understand the baseline performance of classic (non-deep) machine learning models, we trained various regressors on datasets of varying sizes and with different DNA encoding strategies (Fig. 2A). Sequence encoding is needed to featurize nucleotide strings into numerical vectors that can be processed by downstream machine learning models. We considered DNA encodings on three resolutions (Table 1, Fig. 2A): global biophysical properties (Fig. 1A), DNA subsequences (overlapping $k$-mers), and single nucleotide resolution (one-hot encoding).

We trained models on five mutational series chosen because their phenotype distributions are representative of those in the whole dataset (Fig. 1B, Supplementary Fig. S2), and with an increasing number of sequences for training (from ~200 to ~3000 sequences per series). Given the variation in phenotype distributions, we stratified training samples to ensure that their distribution is representative of the full series. We considered four non-deep models: ridge regressor[28] (a type of penalized linear model), multilayer perceptrons[29] (MLP, a shallow neural network with three hidden layers), support vector regressor[30] (SVR, based on linear separation of the feature space with a radial basis function kernel), and random forest regressor[31] (RF, based on axis-aligned splits of the feature space). We chose this array of models because they markedly differ in their principle of operation and underlying assumptions on the shape of the feature space. We tuned model hyperparameters using grid search and 10-fold cross-validation on datasets assembled from aggregated fractions of all mutational series; this allowed us to determine a fixed set of hyperparameters for each of the four models with good performance across the whole dataset (see Methods and Supplementary Fig. S3). In all cases, we assessed predictive accuracy using the coefficient of determination, $R^2$ defined in Eq. (1), between measured and predicted sfGFP fluorescence computed on a set of ~400 test sequences (Supplementary Fig. S3) that were held-out from model training and validation.

In line with expectation, the results in Fig. 2B show that models trained on small datasets are generally poor irrespective of the encoding or regression method. Linear models (ridge) display exceptionally poor accuracy and are insensitive to the size of training set. In contrast, a shallow neural network (multilayer perceptron) achieved substantial gains in accuracy with larger training sets, possibly owing to its ability to capture nonlinear relationships. Our results show that mildly accurate models ($R^2 \geq 50\%$) can be obtained from training sets with ~1000 sequences using random forests and support vector regressors (Fig. 2C). We found random forest regressors to be the most accurate among the considered models, consistently achieving $R^2 \geq 50\%$ for datasets with more than 1000 samples and showing a stable performance when trained on other mutational series (Supplementary Fig. S5). To produce robust performance metrics, the $R^2$ scores in Fig. 2B are averages across five training repeats with resampled training and test sets (Monte Carlo cross-validation).

We also observed a sizeable impact of DNA encodings on prediction accuracy. Subsequence-resolution encodings achieve varying accuracy that is highly dependent on the specific mutational series and chosen model (Fig. 2B, Supplementary Fig. S5). Overall we found a strong preference for base-resolution encodings, with binary one-hot representations achieving the best accuracy. A salient result is that the sequence biophysical properties led to poorer accuracy than most other encodings, possibly due to their inability to describe a high-dimensional sequence space with a relatively small number of features (8). Their poor performance is particularly surprising because the biophysical properties were used to design the sequences based on

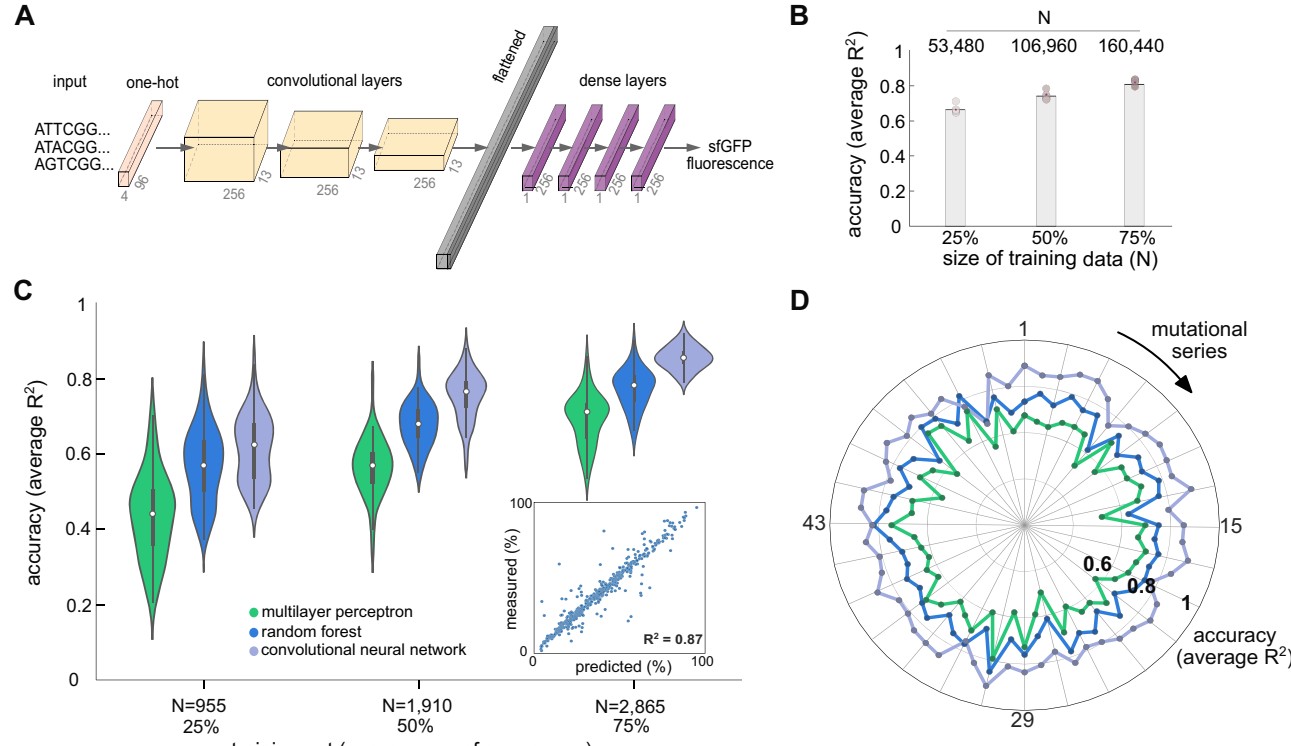

**Fig. 3 | Prediction accuracy of deep neural networks. A** Architecture of the convolutional neural network (CNN) employed in this paper; the output is the predicted sfGFP fluorescence in relative units. The CNN architecture was designed with Bayesian optimization[35] to find a single architecture for all mutational series; our strategy for hyperparameter optimization can be found in the Methods, Supplementary Fig. S3, and Supplementary Tables S4–S5. **B** Accuracy of the CNN in panel A trained on all mutational series. $R^2$ values were computed on held-out sequences (10% of total) and averaged across 5 training repeats; bars denote the mean $R^2$. **C** Prediction accuracy of CNNs against random forest (RF) and multilayer perceptrons (MLPs) on all 56 mutational series using binary one-hot encoding. The CNNs yield more accurate predictions with the same training data. Violin plots show the distribution of 56 $R^2$ values for each model averaged across 5 training

repeats; $R^2$ values were computed on held-out sequences (10% of sequences per series). For all violins, the white circle indicate the median, box edges are at the 25th and 75th percentiles, and whiskers show 95% confidence interval. Inset shows predictions of a CNN trained on 75% of the mutational series with a right-skewed phenotypic distribution (Fig. 1B) computed on held-out test sequences. The CNNs are more complex than the shallow MLPs (2,702,337 vs 58,801 trainable parameters, respectively), but we also found that the CNNs outperform MLPs of comparable complexity (Supplementary Fig. S8); this suggests that improved performance is a result of the convolutional layers acting as a feature extraction mechanism. Details on CNN training can be found in the Methods and Supplementary Fig. S7. **D** Average $R^2$ scores for each model across all 56 mutational series using 75% of sequences for training.

their presumed phenotypic impact[23]; moreover, some of them (codon adaptation index, mRNA secondary structures) represent the state-of-the-art understanding of a sequence impact on translation efficiency[23,32,33], while one-hot encodings lack such mechanistic information. In an attempt to combine the best of both approaches, we trained models on binary one-hot sequences augmented with the biophysical properties ("mixed" encoding in Table 1, Fig. 2B and Supplementary Fig. S5). This strategy led to slight gains in accuracy for small training sets; e.g. for ~200 training sequences, the median $R^2$ with mixed encoding is 0.30 vs a median of 0.26 for binary one-hot (Supplementary Fig. S5). For larger training sets, however, binary one-hot encodings gave the best and most robust accuracy across models.

**Deep learning improves accuracy with the same amount of data**
Prior work has shown that deep learning can produce much more accurate predictions than non-deep models[16,19]. Deep learning models, however, typically require extremely large datasets for training; some of the most powerful deep learning phenotypic predictors, such as DeepBIND[12], Optimus 5-prime[9], ExpressionGAN[34], and Enformer[14] were trained with tens to hundreds of thousands of variants. In the case of sequence-to-expression models, recent literature shows a trend towards more complex and data-intensive models (see Supplementary Table S1); the most recent sequence-to-expression model employed ~20,000,000 promoter sequences to predict protein expression in

*Saccharomyces cerevisiae*[25]. But it is unclear if the accuracy of such predictors results from the model architecture or simply from the sheer size of the training data. To test this idea with our data, we designed a convolutional neural network (CNN, a common type of deep learning model) with an off-the-shelf architecture of similar complexity to those employed in recent literature[25].

Our CNN architecture (Fig. 3A) processes a binary one-hot encoded sequence through three convolutional layers, followed by four dense layers which are equivalent to a four-layer multilayer perceptron. The convolutional layers can be regarded as weight matrices acting on an input sequence. By stacking several convolutional layers, the network can capture interactions between different components of the input. We designed the CNN architecture with a Bayesian optimization algorithm[35] to determine the optimal number of network layers, as well as the optimal settings for the filters in each layer (see Supplementary Tables S4–S5 for details). In addition to the components shown in Fig. 3A, we also included a dropout layer to prevent overfitting and max pooling to reduce the number of trainable parameters. Similar as with the non-deep models in Fig. 2B, hyperparameter optimization was performed by splitting the data into separate sets for training and cross-validation (details in Methods and Supplementary Fig. S3). This allowed us to find a single CNN architecture with good performance across the individual 56 mutational series and the whole dataset.

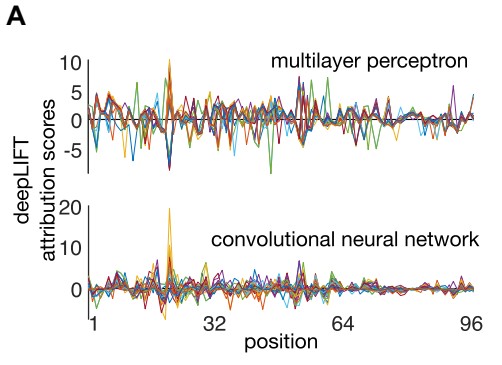
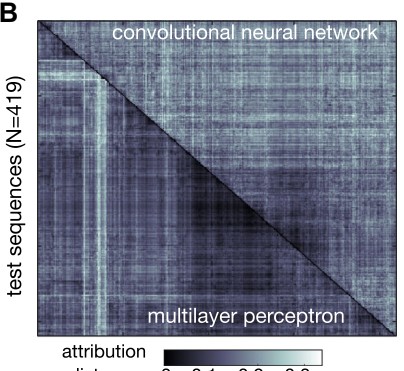
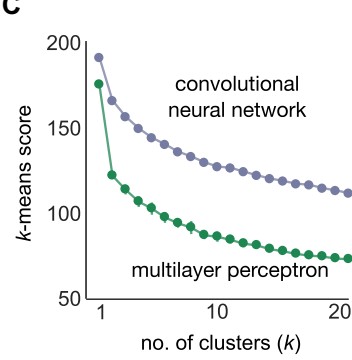

**Fig. 4 | Sensitivity to input sequences using Explainable AI. A** DeepLIFT[37] attribution scores per nucleotide position for a given test sequence and trained model. Panels show scores of 30 sequences chosen at random from the same test set employed in Fig. 3C for models trained on 75% of mutational series 21. **B** Attribution distances for models trained on series 21. We computed the cosine distance between DeepLIFT scores for each sequence in the test set. Distance heatmaps were hierarchically clustered to highlight the cluster structure that both models assign to the input sequences. **C** K-means clustering of the distance matrices in panel B. Line plots show the optimal k-means score averaged across 20 runs with random initial cluster assignments. Lower scores for all values of k suggests that the MLP clusters sequences more heavily than the CNN; we found this pattern in all but four mutational series (Supplementary Fig. S10).

When trained on up to 75% of the full dataset (~160,000 sequences), our CNN model produced excellent predictions in test sets covering broad regions of the sequence space (average $R^2 = 0.82$ across five cross-validation runs, Fig. 3B and Supplementary Fig. S6). This suggests data size alone may be sufficient for training accurate regressors. However, since data of such scale are rarely available in synthetic biology applications, we sought to determine the capacity of CNNs to produce accurate predictions from much smaller datasets than previously considered. To this end, we trained CNNs with the same architecture in Fig. 3A on each mutational series, using ~1000–3000 sequences in each case; details on CNN training can be found in the Methods, Supplementary Fig. S7, and Supplementary Tables S4–S5. We benchmarked the accuracy of the CNNs against non-deep models trained on the same 56 mutational series. As benchmarks we chose two non-deep models: a shallow perceptron (MLP) because it is also a type of neural network, and a random forest regressor because it showed the best performance so far (Fig. 2B). We found that CNNs are consistently more accurate than non-deep models, regardless of the size of the training data (Fig. 3C–D) and across most of the 56 mutational series. In fact, in more than half of mutational series, the CNNs achieve accuracy over 60% with ~1000 training sequences, and in some cases they reach near state-of-the-art accuracy ($R^2 = 0.87$ averaged across five cross-validation runs, Fig. 3C inset). When trained on ~3000 sequences, the CNNs outperformed the MLP in all mutational series, and the random forest regressor in all but four series (Fig. 3D).

To understand why CNNs provide such improved accuracy without larger training data, we performed extensive comparisons against deep MLPs of similar complexity that lack the convolutional layers. We note that the CNNs in Fig. 3C has ~45-fold more trainable parameters than the MLPs, which suggests that such additional complexity may be responsible for the improved predictive accuracy. We thus sought to determine if increasing MLP depth could bring their performance to a level comparable to the CNNs. We trained MLPs with an increasing number of hidden layers on ~3000 sequences from each mutational series. We found that the additional layers provide marginal improvements in accuracy, and that the performance gap between CNNs and MLPs exists even when both have a similar number of trainable parameters (Supplementary Fig. S8). This suggests that the higher accuracy of the convolutional network stems from its inbuilt inductive bias that enables it to capture local structure via the learned filters and more global structure through successive convolutions[36]. As a result, it can capture interactions between different components of the input and produce sequence embeddings that are highly predictive of protein expression.

To further determine how both neural networks process the input sequences, we employed methods from Explainable AI to quantify their sensitivity upon changes in input sequences. We utilized DeepLIFT[37], a computationally efficient method that produces importance scores for each feature of the input; such scores are known as "attribution scores" in the Explainable AI literature[24]. When applied to one-hot encoded sequences, DeepLIFT produces scores at the resolution of single nucleotides (Fig. 4A). We employed these scores to compute pairwise distances between sequences processed by the same model. The shorter that distance, the more the two sequences are detected as similar by the model. We computed such distances for all pairs of sequences in each test set processed by the MLP or CNN. The matrices of pairwise distances (Fig. 4B) were then subjected to hierarchical clustering as a means to contrast the diversity of responses elicited by test sequences on the two models. Using k-means clustering, we showed that the CNN produces less clustered attribution distances than the MLP (Fig. 4C), thus highlighting the ability of the convolutional layers to discriminate input sequences with finer granularity. This trend was found in all but four of the CNNs (Supplementary Fig. S10).

## Impact of sequence diversity on model coverage

A well-recognized caveat of sequence-to-expression models is their limited ability to produce accurate predictions in regions of the sequence space not covered by the training data[25,38]; this is commonly referred to as *generalization performance* in the machine learning jargon. In line with expectation, we found that the CNNs from Fig. 3C, which were trained on a single mutational series each, performed poorly when tested on other mutational series ($R^2 \leq 0$ for most models, Supplementary Fig. S11A); we observed similarly poor results for the non-deep models in Fig. 2 (Supplementary Fig. S11B). Negative $R^2$ scores indicate an inadequate model structure with a poorer fit than a baseline model that predicts the average observed fluorescence for all variants. This means that models trained on a particular region of the sequence space are too specialized, and their phenotypic predictions do not generalize to distant sequences. Although poor generalization can be caused by model overfitting, our cross-validation results (see Supplementary Fig. S6A and Supplementary Fig. S7) rule out this option and suggest that it is rather a consequence of large genotypic differences between mutational series, compounded with the high-dimensionality of the sequence space.

Recent work by Vaishnav and colleagues demonstrated that model generalization can be improved with CNNs of similar complexity to ours[25] trained on large data (~20,000,00 variants). Since the

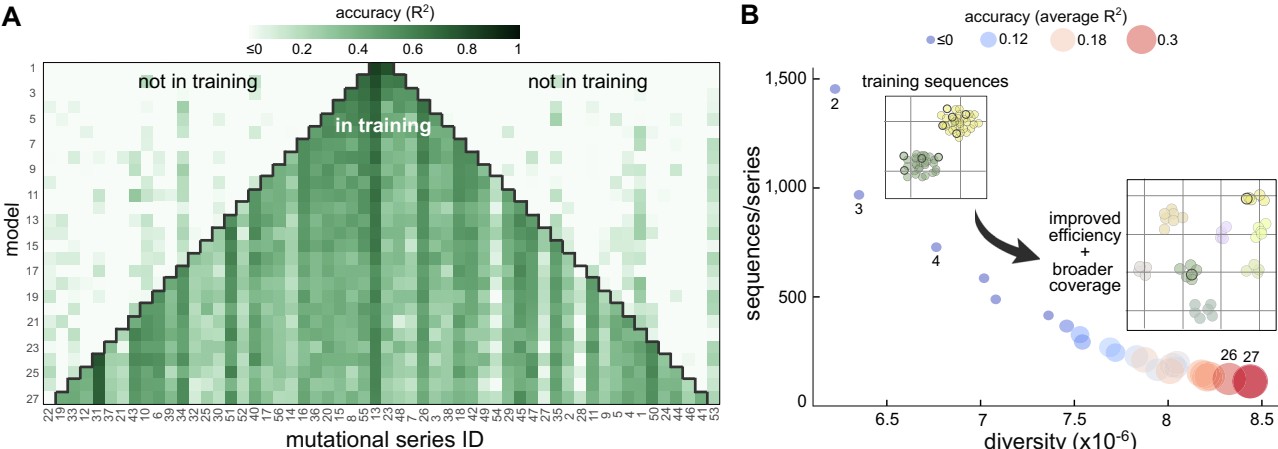

**Fig. 5 | Impact of sequence diversity on data efficiency and model coverage.**
**A** We trained CNNs on datasets of constant size and increasing sequence diversity. We trained a total of 27 models by successively aggregating fractions of randomly chosen mutational series into a new dataset for training; the total size of the training was kept constant at 5800 sequences. Training on aggregated sequences achieves good accuracy for mutational series in the training set, but poor predictions for series not included in the training data. This suggests that CNNs generalize poorly across unseen regions of the sequence space. Accuracy is reported as the $R^2$ computed on 10% held-out sequences from each mutational series. We excluded two series from training to test the generalization performance of the last model. **B** Bubble plot shows the $R^2$ values averaged across all mutational series for each model. Labels indicate the model number from panel A, and insets show schematics of the sequence space employed for training; for clarity, we have omitted model 1 from the plot. Improved sequence diversity leads to gains in predictive accuracy across larger regions of the sequence space; we observed similar trends for other random choices of series included in the training set (Supplementary Fig. S12). The decreasing number of training sequences per series reflects better data efficiency, thanks to an increasingly diverse set of training sequences. To quantify sequence diversity, we counted the occurrence of unique overlapping 5-mers across all sequences of each training set, and defined diversity as $1/\sum_{i=1}^{100} c_i$, where $c_i$ is the count of the $i$-th most frequent 5-mers.

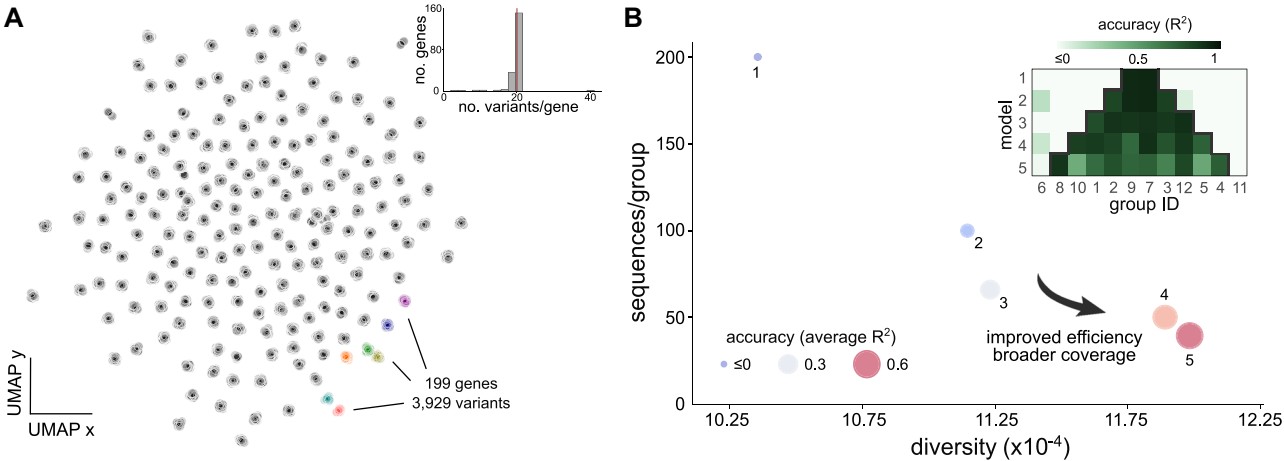

**Fig. 6 | Sequence-to-expression models using promoter data from *Saccharomyces cerevisiae*. A** Genotypic space of yeast promoter data from Vaishnav et al.[25] visualized with the UMAP[27] algorithm for dimensionality reduction; sequences were featurized using counts of overlapping 4-mers, as in Fig. 1B. The dataset contains 3929 promoter variants (80nt long) of 199 native genes, as well as fluorescence measurements of a yellow fluorescent protein (YFP) reporter; inset shows the distribution of variants per gene across the whole dataset. **B** Bubble plots show the accuracy of five random forest (RF) models trained on datasets of constant size and increasing sequence diversity, following a similar strategy as in Fig. 5A. We first aggregated variant clusters into twelve groups, and then trained RF models by aggregating fractions of randomly chosen groups into a new dataset for training; the total size of the training set was kept constant at 400 sequences. Accuracy was quantified with the $R^2$ score averaged across test sets from each group (~30 sequences/group) that were held out from training. Inset shows model accuracy in each test set. In line with the results in Fig. 5A, we observe that model coverage can be improved by adding small fractions of each group into the training set; we observed similar trends for other random choices of groups included in the training set (Supplementary Fig. S13). Details on data processing and model training can be found in the Methods and Supplementary Text. Sequence diversity was quantified as in Fig. 5B.

cost of such large screens is prohibitive in most synthetic biology applications, we sought to understand how model coverage could be improved in scenarios where data size is strongly limited. The idea is to design a sequence space for training that can enlarge the high-confidence regions of the predictors with a modest number of variants. To this end, we performed a computational experiment designed to test the impact of sequence diversity on the ability of CNNs to produce accurate predictions across different mutational series.

We trained CNNs on datasets of constant size but increasing sequence diversity (Fig. 5A, Supplementary Fig. S12). We considered an initial model trained on 5800 sequences sampled from the aggregate of two series chosen at random, e.g. 2900 sequences from series 13 and 23, respectively (Fig. 5A top row). We successively added two

series to the aggregate and retrained a CNN while keeping a constant number of total sequences. This results in sparser sampling from each mutational series and an increasingly diverse training set. For example, the second model (Fig. 5A) was trained on 1450 sequences from series 13, 23, 48 and 55, respectively. Overall, we trained a total of 27 models, the last of which comprises as few as 107 sequences per mutational series. The resulting models display substantial variations in their predictive power (Fig. 5A). Most models displayed variable $R^2$ scores across different series, and we identified two salient patterns: some series that are consistently well predicted even in small data scenarios (e.g. series 31 and 51), and some series are particularly hard to regress (e.g. series 28 and 54), which possibly require a bespoke CNN architecture different from the one in Fig. 3A. The results also show that increased diversity has a minor impact on model generalization; although some series not included in training do have improved prediction scores (e.g. series 53 in Fig. 5A), we suspect this is likely a result of series being particularly easy to regress. In general, we observed patterns of low or negative $R^2$ scores for series not included in the aggregate. Similar results were observed for other random choices of mutational series employed for training (Supplementary Fig. S12).

Crucially, the results in Fig. 5B suggest that increased sequence diversity enlarges the region where the CNN can produce accurate predictions without increasing the size of the training data. We found that $R^2 > 30\%$ in many regions of the sequence space can be achieved by models trained on just over a hundred sequences from those regions (e.g. model 27 in Fig. 5A). For comparison, the CNN trained on all series without controlled diversity can double that accuracy, but with a 9-fold increase in the size of the training data ($R^2 = 0.65$ for $N = 53{,}480$ in Fig. 3B). This means that model coverage can be enlarged with shallow sampling of previously unseen regions of the sequence space, which provides a useful guideline for experimental design of screens aimed at training models on a limited number of variants.

To test the validity of this principle in a different expression chassis and construct library, we repeated the analysis in Fig. 5 using a recent genotype-phenotype screen of promoter sequences in *Saccharomyces cerevisiae*[25]. These data are comparable to the screen in Cambray et al.[23] in the sequence length (80nt) and its highly clustered coverage of genotypic space (Fig. 6A). This clustered structure results from the design of the library itself, which is composed of 3929 variants of 199 natural promoters. A key difference between this new dataset and Cambray et al.[23] is the construct architecture; unlike the UTR sequences in Fig. 1B, promoter sequences account for regulatory effects but do not undergo transcription. Akin to our results in Fig. 5, we aimed at testing the accuracy of machine learning regressors trained on datasets of constant size but increasing sequence diversity. Since this dataset contains a small number of variants for each gene (on average 20 variants/gene, see inset of Fig. 6A), we first randomly aggregated the variant clusters into twelve groups containing an average of 327 sequences/group. We subsequently trained five Random Forest models on $N = 400$ binary one-hot encoded sequences drawn from different groups. For example, as shown in the Fig. 6B, model 1 was trained on 200 sequences from two groups, whereas model 2 was trained on 100 variants from four groups. The training results (Fig. 6B) show a strikingly similar pattern to those observed in our original dataset in Fig. 5, thus strongly suggesting that sequence diversity can be exploited to train models with broader coverage and improved data efficiency.

## Discussion

Progress in high-throughput methods has led to large improvements in the size and coverage of genotype-phenotype screens, fuelling an increased interest in deep learning algorithms for phenotypic prediction[9,12–14,16,17,19,34]. Synthetic biology offers a host of applications that would benefit from such predictors, e.g. for optimization of protein-producing strains[39], selection of enzymatic genes in metabolic

engineering[40], or the design of biosensors[41]. An often-overlooked limitation, however, is that deep learning models require huge amounts of data for training, and the sheer cost of the associated experimental work is a significant barrier for most laboratories. Recent sequence-to-expression models have focused primarily on datasets with tens to hundreds of thousands of training sequences (Supplementary Table S1). While large data requirements are to be expected for prediction from long sequences such as entire protein coding regions, synthetic biologists often work with much shorter sequences to control protein expression levels (e.g. promoters[3], ribosomal binding sequences[4], terminators[42] and others). From a machine learning standpoint, shorter sequences offer potential for training models with smaller datasets, which can lower the entry barriers for practitioners to adopt deep learning for strain optimization.

Here, we examined a large panel of machine learning models, with particular emphasis on the relation between prediction accuracy and data efficiency. We used data from an experimental screen in which sequence features were manipulated using a Design of Experiments approach to perturb the translation efficiency of an sfGFP reporter in *E. coli*[23]. Thousands of local mutations were derived from more than fifty sequence seeds, yielding mutational series that enable deep focal coverage in distinct areas of the sequence space (Fig. 1B). By suitable sampling of these data, we studied the impact of the size and diversity of training sequences on the quality of the resulting machine learning models.

Our analysis revealed two key results that can help incentivize the adoption of machine and deep learning in strain engineering. First, in our dataset we found that the number of training sequences required for accurate prediction is much smaller than what has been shown in the literature so far[8,12,16,17,25]. Traditional non-deep models can achieve good accuracy with as few as 1000–2000 sequences for training (Fig. 2B). We moreover showed that deep learning models can further improve accuracy with the same amount of data. For example, our convolutional neural networks achieved gains of up to 10% in median prediction scores across all mutational series when trained on the same 2000 sequences as the non-deep models (Fig. 3C). Such performance improvement is a conservative lower bound, because we employed a fixed network architecture for all mutational series; further gains in accuracy can be obtained with custom architectures for different mutational series.

Second, we found that sequence diversity can be exploited to increase data efficiency and enlarge the sequence space where models produce reliable predictions. Using two different datasets with a similar structure of their sequence coverage, the *E. coli* library from Cambray et al.[23] as well as a recently published library of *S. cerevisiae* promoters[25], we showed that machine learning models can expand their predictions to entirely new regions of the sequence space by training on a few additional samples from that region (Figs. 5, 6). This means that controlled sequence diversity can improve the coverage of sequence-to-expression models without the need for more training data. In other words, instead of utilizing fully randomized libraries for training[8,16–18], it may be beneficial to first design few isolated variants for coverage, and then increase the depth with many local variants in the vicinity of each seed. Our work strongly suggests that such balance between coverage and depth can be advantageous in small data scenarios, where fully randomized libraries would lead to datasets with faraway and isolated sequences that inherently require large datasets to achieve high accuracy. This principle is conceptually akin to the idea of "informed training sets" introduced by Wittmann and colleagues[43] in the context of protein design, which can provide important benefits in cases where data efficiency is a concern. Our observations raise exciting prospects for Design of Experiments strategies for training predictors of protein expression that are both accurate and data-efficient.

Data requirements above 1000 sequences are still too costly for most practical applications. Further work is thus required on DNA

encodings that are maximally informative for training, as well as model architectures that can deliver high accuracy for small datasets. Both strategies have proven highly successful in protein engineering[44,45], yet their potential for DNA sequence design remains largely untapped. We found that seemingly superficial changes to DNA encodings, e.g. from binary one-hot to ordinal one-hot encodings (Fig. 2B), can have substantial impact on predictive performance. Moreover, although biophysical properties such as the CAI or the stability of mRNA secondary structures are not good predictors by themselves[17], we observed small but encouraging improvements when these were employed in conjunction with one-hot encodings, particularly for small datasets. This suggests that richer mechanistic descriptors, e.g. by including positional information or base-resolution pairing probabilities of secondary structures, may yield further gains in accuracy.

In agreement with other works[46], we observed that sequence-to-expression models generalize poorly: their accuracy drops significantly for sequences that diverge from those employed for training. This limitation is particularly relevant for strain engineering, where designers may employ predictors to navigate the sequence space beyond the coverage of the training data. A recent study by Vaishnav et al. illustrated that these models can indeed generalize well using a massive training set with over 20,000,000 sequences[25]. Data of such scale are far beyond the capacity of most laboratories, and therefore it appears that poor generalization is likely to become the key limiting factor in the field. We suggest that careful design of training libraries in conjunction with algorithms for controlled sequence design[38] may help to improve sequence coverage and avoid low-confidence regions of the predictors.

Deep learning models promise to deliver large gains in efficiency across a range of synthetic biology applications. Such models inevitably require training data and there is a risk that the associated experimental costs become an obstacle for many laboratories. In this work we have systematically mapped the relation between data size, diversity and the choice of machine learning models. Our results demonstrate the viability of more data-efficient deep learning models, helping to promote their adoption as a platform technology in microbial engineering.

## Methods

### Data processing

**Data sources and visualization.** The *E. coli* dataset presented by Cambray et al.[23] was obtained from the OpenScience Framework[47]. After removing sequences with missing values for sfGFP fluorescence and growth rate, the dataset contains ~228,000 sequences. In all trained models, we employed the arithmetic mean of sfGFP fluorescence across replicates for the case of normal translational initiation[23]. To visualize sequences in a two dimensional space (Fig. 1B), we employed the UMAP algorithm[27] v0.5.1 on sequences featurized on counts of overlapping $k$-mers. We found that the UMAP projection improved for larger $k$, and chose $k = 4$ to achieve a good trade-off between computation time and quality of projection (Supplementary Fig. S1); $k$-mer counting was done with custom Python scripts. In all cases, fluorescence measurements were normalized to the maximum sfGFP fluorescence across cells transformed with the same construct averaged over 4 experimental replicates of the whole library[23].

**Training, validation, and test data.** In Supplementary Fig. S3A we illustrate our strategy to partition the full dataset into sets for training, cross-validation and model testing. For each mutational series, we first perform a split retaining 10% of sequences as a fixed held-out set for model testing. We use the remaining sequences as a development set and perform a second split to obtain two partitions for each series. The first partition is for model training and comprises 3200 sequences from which we used varying fractions for training regressors in each series. The second partition was employed for hyperparameter optimization, containing ~400 sequences from each series (10% of the whole series) that we then merged into a large validation set comprising

22,400 sequences (56 series × 400 sequences per series) from all series. We kept the validation set fixed and employed it for hyperparameter optimization of both non-deep and deep models. In all data splits, we stratified the sfGFP fluorescence data to ensure that the phenotype distributions are preserved. Stratification was done with the verstack package, which employs binning for continuous variables; we further customized the code to gain control of the binning resolution.

### Model training

**Non-deep machine learning models.** DNA encodings (Table 1) were implemented with custom Python code, and all non-deep models were trained using the scikit-learn Python package. To determine model hyperparameters, we used a validation set for all combinations of encodings and regressors. As illustrated in Supplementary Fig. S3B, for each model we explored each the hyperparameter search space (Supplementary Table S3) for all encodings using grid search with 10-fold cross-validation on 90% of our validation set (~20,000 sequences), using mean squared error (MSE) as performance metric. This resulted in six hyperparameter configurations for each regressor (one for each encoding). For many regressors, we found that the same configuration was optimal for several encodings simultaneously, and we thus settled on most frequent configuration among the six encodings; in case of a tie between configurations, we settled for the one with the best MSE computed on the remaining 10% of our whole validation set.

**Convolutional neural networks.** CNNs were trained on Tesla K80 GPUs from Google Colaboratory. To design the CNN architectures, we use the Sequential class of the Keras package with the TensorFlow backend[48,49]. All CNNs were trained on binary one-hot encoded sequences with mean squared error as loss function, batch size of 64, learning rate $1 \times 10^{-3}$, and using the Adam optimizer[50]. Since Adam computes adaptive learning rates for each weight of the neural network, we found that the default options were adequate and did not specify a learning rate schedule. We set the maximum number of epochs to 100, and used 15 epochs without loss improvement over the validation set as early stopping criterion to prevent overfitting.

Model hyperparameters were selected with Bayesian optimization implemented in the HyperOpt package[35]. Specifically, as shown in Supplementary Figure S3C, we performed five iterations of the HyperOpt routine using 90% of our validation set (~20,000 sequences), where subsets of the search space were evaluated (Supplementary Table S4). We used the Tree of Parzen Estimators (TPE)[51] as acquisition function, and set the number of architecture combinations to 50. This resulted in five candidate architectures, from which we chose the one with the best validation MSE computed on a stratified sample of size 10% of the whole validation set. The resulting model architecture is described in Supplementary Table S6. To verify that the selected architecture works best for our study, we performed an additional test (Supplementary Fig. S9) where we trained CNNs of varying width and depth and compared them to the results in Fig. 3C. To achieve this, we perturbed the number of convolutional filters and layers, for width and depth respectively, and trained the resulting architectures using 75% of sequences for each mutational series (Supplementary Fig. S9).

**Model testing.** In all cases we did five training repeats on resampled training sets and a fixed test set. Model accuracy was computed as coefficient of determination ($R^2$) on held-out sequences, averaged across five training repeats. The $R^2$ score for each training repeat was defined as:

$$R^2 = 1 - \frac{\sum_i (y_i - f_i)^2}{\sum_i (y_i - \bar{y})^2}, \tag{1}$$

where $y_i$ and $f_i$ are the measured and predicted fluorescence of the $i^{\text{th}}$ sequence in the test set, respectively, and $\bar{y}$ is the average fluorescence

across the whole test set. Note that for a perfect fit we have $R^2 = 1$, and conversely $R^2 = 0$ for baseline model that predicts the average fluorescence (i.e. $f_i = \bar{y}$ for all sequences). Negative $R^2$ scores thus indicate an inadequate model structure with worse predictions than the baseline model.

### Interpretability analysis

For the interpretability results in Fig. 4A–C, we employed DeepLIFT[37] which utilizes back-propagation to produce importance or "attribution" scores for input features, with respect to a baseline reference input. We chose a blank sequence as a reference. We used the *GenomicsDefault* option that implements *Rescale* and *RevealCancel* rules for convolutional and dense layers, respectively. The line plots in Fig. 4A are the attribution scores of 30 random test sequences for the CNN and MLP models trained on mutational series 21. The distance heatmaps in Fig. 4B were produced by computing the cosine distance between vectors of attribution scores, and then using hierarchical clustering to compare both models. The degree of clustering was quantified by *k*-means scores (Fig. 4C); lower scores suggest more clustering of the distance matrix. Results for all other mutational series can be found in Supplementary Figure S10.

### Impact of sequence diversity

***Escherichia coli* dataset**. The models in Fig. 5 were trained on data of constant size and increasing sequence diversity. We successively aggregated fractions of mutational series to create new training sets with improved diversity. We employed the same CNN architecture and training strategy as in Fig. 3A with the same hyperparameters (Supplementary Table S6) for all 27 models. To ensure a comparison solely on the basis of diversity, we fixed the size of the training set to 5800 sequences. To increase diversity, for successive models we sampled training sequences from two additional series, as shown in Fig. 5. The specific series for the aggregates were randomly chosen; four training repeats with randomized selection of series can be found in Supplementary Figure S12.

***Saccharomyces cerevisiae* dataset**. We obtained the promoter dataset presented in Supplementary Fig. 4F in Vaishnav et al.[25] from CodeOcean[52]. The data contains 3929 yeast promoter sequences with YFP fluorescence readouts. To visualize the yeast sequences (Fig. 6A), we employed the same strategy as in Fig. 1B for the *E. coli* dataset, and used the UMAP algorithm for counts of overlapping 4-mers. Additional details can be found in the Supplementary Text.

For the models in Fig. 6B, we first aggregated sequences from the clusters in Fig. 6A into twelve groups. We then employed the same strategy as in Fig. 5, and successively aggregated fractions of groups to create new training sets with improved diversity. We used the same Random Forest configuration (Supplementary Table S6) for all 5 models. We fixed the size of the training set to 400 sequences, and to increase diversity for successive models, we sampled training sequences from two additional groups at a time (Fig. 6B). The specific groups for the aggregates were randomly chosen; four training repeats with randomized selection of groups can be found in Supplementary Fig. S13. Additional details can be found in the Supplementary Text.

### Reporting summary

Further information on research design is available in the Nature Portfolio Reporting Summary linked to this article.

### Data availability

The genotype-phenotype data employed in this study come from two literature sources[23,25]. For reproducibility, both datasets have been cleaned and reorganized in a form suitable for machine learning analyses; the cleaned data used in this study are available in Zenodo[53] at https://doi.org/10.5281/zenodo.7273952.

### Code availability

Python code for model training, data analysis, and data plotting can be run from Google Colaboratory and is available in Zenodo[53] at https://doi.org/10.5281/zenodo.7273952.

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

## Acknowledgements
E.M.N. was supported by a doctoral studentship from the Darwin Trust of Edinburgh. D.A.O. was supported by the United Kingdom Research and Innovation (grant EP/S02431X/1).

## Author contributions
E.M.N. and D.A.O. designed the research and analyzed data. E.M.N. performed model implementation and training. A.W. tested code implementation and provided general feedback. G.C. and O.M.A. provided counsel on data analysis and computational aspects. D.A.O. provided overall supervision and direction of the work.

## Competing interests
The authors declare no competing interests.
