## [Peer Review File · Nature Communications]

Reviewers' Comments:

Reviewer #1:

Remarks to the Author:

The authors make use of a previously published data set of one of the co-authors in order to test different machine learning approaches and compare critical parameters such as dataset size and diversity as well as model architectures (classical vs. deep learning). The goal is to predict gene expression (sfGFP) directly from sequence.

While this is a useful effort to systematically assess different models and conditions, I have trouble pointing to the novelty or surprising findings in this study. Many of the major claims have been shown elsewhere. Some examples:

- The fact that model accuracy rises with increasing training diversity is common sense and has already been shown by several studies
- Similar argument may be made for increasing training set size as well as for the benefits of deep learning over conventional ML (the authors may wish to carefully look for instance into reference 16; the ML part in this paper shows these things for a CNN in almost the same context, i.e. translation in bacteria)
- The fact that the choice of encodings is critical is by no means new

Further, the authors should carefully revise some of their claims on novelty and generalizability. E.g.:

- the fact that deep learning is data intense is widely known
- The claim that the authors have established that "much smaller datasets than previously thought" can produce accurate predictions is simply false. Other groups (e.g. Arnold or Church amongst many others) have shown that for much fewer sequences
- Crucially, I believe that the generalizability of some of the quantitative claims is questionable. As an example, saying that "traditional non-deep models can achieve [...] $R^2 > 70\%$ with as few as 2,000 sequences" may be correct for the underlying dataset, problem and model, but it is highly unlikely that these numbers have any general meaning (they will be different for promoters, proteins, organisms etc.).

Again, I believe that some of the systematic analyses are helpful and of interest for the community. However, given the points above I would recommend publication of this work in a more specialized journal after careful revision.

General remarks:

- Drawbacks of sequence design: The authors nicely point out the advantages of the way how they (or at least one coauthor) designed the sequence space in the library in their previous paper (Cambray et al.); However, designing libraries based on certain prior hypotheses also bares a risk: the introduction of user-borne bias and overfitting to the design. That should be critically discussed, especially since the authors see such effects (e.g. lines 156-161).
- To the same end, the selection of test sets is highly critical. Since the test set follows the same modular design principles as the training data, there is a substantial risk that the models simply learns to recapitulate the design rules imposed by the user. E.g. it might detect certain sequence motifs that occur repeatedly as a result of the seed-based design approach. I believe the fact that the models are incapable to generalize across the different series could speak for this. At least the authors should test for that, e.g. by testing the models on held-out test sets that do not follow any design (randomness has its advantages there).
- How did the authors ensure that no overfitting is happening? I could not find corresponding analyses (e.g. the validation R^2 s etc.)
- The panels C and D in Fig. 3 are not discussed in the main text, which leaves a gap in the story.
- The methods section is very slim, especially on ML, considering that the authors have tested "thousands of models". I have doubts that this is presented in a way that would facilitate repetition or further use by the community in one page. A detailed ML description should be provided.

Specific comments:

- Line 48: "comprehensive" seems like an overclaim

- Line 67: it is not clear why “random sequences obscure the effect of few specific mutations”, the meaning here should be clarified; if I understand correctly, I would argue that what the authors mean is not a result of the fact that a library is random but that it’s hard to achieve sufficient coverage for fully random libraries
- Fig. 1C: It does not seem like a big surprise that there are five different clusters showing different distribution shapes in the phenotype (GFP measurement; side note: it’s not clear what 100% refers to, the maximum in the cluster or the entire data); How were those five selected? More importantly, it is not clear how that observation makes for an “ideally suited” dataset for ML. The authors should back the corresponding claim by a more detailed explanation and by showing the phenotypic distribution in the entire library (not five cherry picked clusters).
- Fig. caption of Fig. 2, statement on test set: It’s not 100% clear whether there is one test set or one for each series; Also “10% from each series”, does that mean from each of the five series that were used for training, or from all 56 series? If the former is the case, there is a risk that the models perform well on the selected series but much less accurate on other series let alone on sequences that do not follow any of the design criteria in the library (see above comment on the risk of pre-designed libraries!)
- Fig. 2C: the axes labels “predicted” and “measured” are not descriptive; what is the percentage (what are these values normalized to)?
- Line 136-137: “This suggest... regressors”: This phrase is meaningless without providing quantitative information, e.g. on what the authors consider as “large enough”.
- Line 158: It’s unclear how R^2 can become negative
- 198-199: the fact that deep learning requires a lot of data is by no means “often overlooked” but widely accepted and known
- Lines 214-215: The authors should specify what “the literature suggest” and add corresponding references. I would argue that there is no common-sense minimal number of sequences that is required for deep learning (so the claim is false)
- Line 250-251: The phrase “comprehensively mapped” seems exaggerated given the sheer infinite number of combinations of dataset size, diversities and models that one could test. “systematically” would be a fair term imo.

Reviewer #2:

Remarks to the Author:

****Summary:****

- This study presents a comprehensive study of deep (Convolutional Neural Networks) and non-deep learning (Ridge regression, Multilayer Perceptron, Support Vector Regression, and Random Forest) models in protein expression. This work pays special attention to the data efficiency aspect of these machine learning models. The results show that deep learning models require less labelled data than usually expected from the literature.

****Contributions:****

- The paper does a comprehensive study of several machine learning models applied to the problem of predicting protein expression using a dataset acquired from Escherichia coli.

- The paper shows that deep learning models might not need as large amounts of labelled information as currently stated in the literature.

****Strengths:****~

- This manuscript is well written and easy to follow.

- This study provides an interesting benchmark of several machine learning models for the problem of predicting protein expression.

****Weaknesses:****

The manuscript is interesting, but some crucial question pertaining to the main claim and contribution of the paper should be addressed before publication:

1. How are the hyper-parameters of all these machine learning models optimized? The right practice in machine learning is to split the data as a development and test set, where the development set is itself split into training and validation sets. The training set is used to train the model itself, for example the weights of an MLP, while the validation set is used to optimize the hyper-parameters of the model. Finally, the model is evaluated using the test set with the model trained with the best hyper-parameters from the validation set. From Section III B, we can notice that hyper-parameters "were determined via grid search and 10-fold cross validation". But, what data is used to construct the 10-folds? Is the performance of the hyper-parameter search computed using the test data? This is especially critical in deep learning models that are very prone to overfitting. If not done correctly, this issue could invalidate the main conclusion of this paper about the effectiveness of deep learning methods in low data regimes.

2. What are the expected results with a deep MLP? Indeed the final part of the CNN is an MLP of four layers. The convolutional layers have been used in the literature as a feature extraction mechanism, which has some benefits like rotation and translation invariance. It would be interesting to know how a deep MLP compares with the CNN, i.e., considering both models with a comparable number of learnable parameters.

3. How the mixed representation is constructed in Section I B is confusing. If I understood well, the physical representation is given by an 8-dimensional vector for the whole sequence, while the one-hot encoding is a matrix of dimension $4 \times L$, where L is the dimension of the sequence. Is the biophysical vector concatenated to each amino-acid, resulting in a matrix of dimension $12 \times L$? Otherwise, what is the strategy adopted in this case?

4. Reference to different CNNs is confusing. For example in this sentence "This trend was found in all but four of the CNNs". I believe all the CNNs are the same, but trained with different amounts of data (25%, 50%, 75%). However, it is unclear what the authors refer to when discussing different CNNs. Are there more CNN architectures beside the CNN in Fig. 3A? Please clarify.

5. It might not be surprising that the proposed CNN model can perform well with few labelled information. First, this CNN model is not very complex. From Fig. 3A we can see that it has 3 convolutional layers, plus 4 fully connected layers. This study should report the number of learnable parameters for the CNN and the MLP. Second, we know from other fields such as Computer Vision that deep CNN with millions of parameters perform well with the ImageNet dataset, which contains around one million data points. I believe the ratio between data points and number of parameters in this paper should be similar to that in the ImageNet problem. There is a new emerging theory in machine learning about double descent for over-parametrized models such as CNNs. This new theory tries to explain the good performance of CNN models for low data regimes. In that sense, I believe a more interesting question is to establish how shallow or deep a CNN should be to perform well with the dataset considered in this paper. To that end, this study should perform more experiments with CNN of different complexities, maybe something like: current model, 8 convolutional layers CNN, 16 convolutional layers CNN, and 32 convolutional layers CNN. I believe this would yield a very interesting discussion.

6. Do the classical machine learning models (Ridge regression, MLP, SVR, RF) also have a poor performance in generalization? If simpler models like RF do not have generalization issues, the problem observed with the CNN could be a problem of overfitting. Has the hyperparameters optimization been done properly (please see comment 1)?

7. I hesitate to accept the main conclusion of this paper about CNNs requiring less labelled data than suggested in the literature. From one point of view, the Vapnik–Chervonenkis (VC) dimension in classical machine learning theory gives us some bounds in terms of the sample complexity required to solve a classification problem. We can roughly approximate the VC dimension of a neural network as the number of learnable parameters. A rule of thumb, used before the deep learning era, was to have ten times the VC dimension of data to avoid overfitting. I believe the dataset in this paper does not abide to the sample complexity given by the VC dimension for the

CNN model. From another point of view, we now have the double descent theory of over-parametrized models, but having only one CNN model makes it impossible to draw some conclusions. I suggest the authors thoroughly check this affirmation, which is perhaps the most important contribution of this paper. Please also check if the other models suffer from the stated generalization issue.

****Decision:****

This manuscript presents an interesting discussion about deep and non-deep learning models for protein expression. The main claim of this paper is that deep learning models do not require large amounts of data to perform well. However, more experiments should be done to confirm this claim. I suggest Major Revisions, and I encourage the authors to thoroughly review the paper to better support the main claim of the paper.

****Typos:****

Line 24 "high-throughout" → high-throughput

Response to reviewers

“Accuracy and data efficiency in machine learning models of protein expression”

We thank reviewers for their comments. We have addressed them in the point-by-point response attached. The revised manuscript is a substantial improvement and **includes 10 new figures, 4 new tables, a new dataset, and a new Supplementary Text**. We have also added 8 new references and revised large portions of the main text. Table 1 below summarizes the new materials and all changes have been tagged in blue in the revision.

In addition to the changes suggested by reviewers, we have included new results that will broaden the impact of our manuscript. Specifically, we validated our original sequence diversity results from *Escherichia coli* (Figure 4) with a new dataset by Vaishnav et al, Nature, 2022 containing promoter sequences in *Saccharomyces cerevisiae*. The new results in Figure 5 confirm our original finding: controlled sequence diversity can improve the coverage of expression models without the need for more data. The results in Figure 5 thus extend such findings to a different organism and a different construct library.

Item	Description	Referee
Suppl Fig S2	Phenotypic distributions of all mutational series in Cambray et al, Nat Biotech, 2018.	1
Suppl Fig S3	Design of training, validation and test sets, plus strategies for hyperparameter optimization.	both
Suppl Fig S4	Cross-validation of non-deep models in Figure 2B.	both
Suppl Fig S6A	Training and validation loss of CNN models trained on the full dataset from Cambray et al, Nat Biotech, 2018.	both
Suppl Fig S7	Training and validation loss of CNN models trained on each of the 56 mutational series from Cambray et al, Nat Biotech, 2018.	both
Suppl Fig S8	Comparison of MLPs of increasing depth with CNNs.	2
Suppl Fig S9	CNNs with variable no. of convolutional layers and filter width.	2
Suppl Fig S11	Generalization performance of CNNs and non-deep models.	both
Suppl Table S1	Sequence-to-expression models built so far in the literature, highlighting the novelty of our focus on small datasets.	1
Suppl Table S2	Search space for hyperparameters of non-deep models in Figure 2.	both
Suppl Table S4	Search space for hyperparameters of the CNN in Figure 3A.	both
Figure 5	New results to validate conclusions of Figure 4 in a separate dataset from Vaishnav et al, Nature, 2022.	n/a
Suppl Fig S13	Randomized repeats of the computational experiment in Figure 5	n/a
Suppl Table S6	Search space for hyperparameters of the RFs in Figure 5.	n/a
Suppl Text	Describes the data employed for the new results in Figure 5.	n/a
Suppl Data	3,929 promoter sequences from Vaishnav et al, Nature, 2022. Data has been cleaned and organized for ease of reproducibility.	n/a

Table 1. Summary of new material in the revision.

Point-by-point response

Reviewer #1

The authors make use of a previously published data set of one of the co-authors to test different machine learning approaches and compare critical parameters such as dataset size and diversity as well as model architectures (classical vs. deep learning). The goal is to predict gene expression (sfGFP) directly from sequence.

While this is a useful effort to systematically assess different models and conditions, I have trouble pointing to the novelty or surprising findings in this study. Many of the major claims have been shown elsewhere. Some examples:

We thank the reviewer for the description of our work. Below we address each point in detail.

Q: *The fact that model accuracy rises with increasing training diversity is common sense and has already been shown by several studies.*

A: Without literature references it is difficult to know what studies the reviewer is referring to. We are not aware of studies on the relation between DNA sequence diversity and machine learning models of protein expression. Moreover, while we agree that the claim ‘*model accuracy rises with increasing training diversity*’ may sound obvious from a general machine learning standpoint, we suspect there is a misunderstanding here because this is not what we claim in our manuscript.

Specifically, our results in Figure 4 suggest something more nuanced: *in small data scenarios, sequence diversity can improve performance with the same data size*. In mainstream applications of machine learning, particularly those where data is comparatively cheap, increased data diversity also amounts to *more* data for training. But sequence-to-expression data is costly and makes many deep learning models prohibitively expensive to train. **Our central premise is that data size is a key barrier for the adoption of deep learning in synthetic biology**, and our study is the first to deliberately focus on sequence-to-expression models with small data.

In the new Figure 5, we have further validated our conclusion on sequence diversity using a recently published dataset by Vaishnav et al, Nature, 2022. The data contains 3,929 promoter sequences from *Saccharomyces cerevisiae* and was originally employed as a test set for their expression model, see Supplementary Figure 4F of Vaishnav et al. We repurposed these data as a training set for our models to assess the validity of our conclusion in a different cellular host and a different construct library. We are confident these new results will substantially improve the impact of our manuscript in the synthetic biology community.

In other words, our results on sequence diversity (Figures 4-5) say that if we have a limited budget for phenotyping N strains, instead of designing a fully randomized library, it is better to design few isolated seed variants for coverage, and then increase depth with local variants in the vicinity of each seed. We suggest that such balance between coverage and depth is beneficial in low-N scenarios, where fully randomized libraries would lead to datasets with distant and isolated sequences that would inherently require very large datasets for training accurate models. The work in Vaishnav et al is an excellent example of the latter point: their model was trained on a fully randomized library, but more than 20,000,000 sequences were required for good accuracy.

We have included a new paragraph on these points in the Introduction (L41-50) and Discussion (L325-341).

Q: *Similar argument may be made for increasing training set size as well as for the benefits of deep learning over conventional ML (the authors may wish to carefully look for instance into reference 16; the ML part in this paper shows these things for a CNN in almost the same context, i.e. translation in bacteria)*

A: We agree that the relation between performance and data size is well known, and we did not claim this as a result. In line with expectation, in Figure 2B we indeed observed an increase in performance for larger datasets, but the text in Section Results (A) aims to give the reader baseline performances for datasets of realistic size (from N=200 to N=2,800 sequences); we do not highlight the increase in performance as a result in itself.

With regards to the benefits of deep learning over classic models, we agree that this is generally expected *if data size is not taken into consideration*. What is not obvious, is that in our data set, deep learning improves performance *without the need for more training data*. This is the message of Figure 3C-D.

With regards to the work in Höllerer et al, Nat Commun, 2020, we highlight that they trained models on datasets of increasing size, from N=2,500 up to N=248,000 sequences; see Figure 4E in Höllerer et al, Nat Commun, 2020. Our work instead explores data sizes in the opposite direction, with our largest data set (N=2,800) being comparable to the smallest one in Höllerer et al. Therefore, our study is fundamentally different in that it deliberately focuses on small data sizes. Table 2 below lists the papers that address the same biological question as ours (prediction of protein expression from DNA sequence), which clearly demonstrates the novelty of our focus on small data. We have included this table as Supplementary Table S1 in the revision, so as to highlight the contribution of our work with respect to the state-of-the art.

Reference	Length	Data size
de Boer et al, Nat Biotech, 2020	80nt	100,000,000
Vaishnav et al, Nature, 2022	80nt	20,000,000
Kotopka et al, Nat Commun, 2020	246-312nt	1,000,000
Cuperus et al, Genome Res, 2017	50nt	500,000
Angenent-Mari et al, Nat Commun, 2020	145nt	90,000
Höllerer et al, Nat Commun, 2020	17nt	2,500-248,000
Our study	96nt	190-2,800

Table 2. Recent sequence-to-expression machine learning models from the literature. The list focuses on studies on short DNA or RNA sequences. We exclude studies on whole gene prediction (e.g. Avsec et al, Nat Methods, 2021) or on prediction of other phenotypes beyond protein expression, such as transcription factor binding (e.g. Aliphani et al, Nat Biotech 2015) or mRNA levels (e.g. Agarwal & Shendure, Cell Reports, 2020).

Q: *The fact that the choice of encodings is critical is by no means new*

A: We did not claim the impact of encodings as a novel result. What we actually found are biologically meaningful conclusions that add to the growing body of literature on the genetic determinants of protein expression:

- 1) We found that common sequence properties such as the Codon Adaptation Index, AT content, and minimum free energy of secondary structures (see Figure 1A) are poor predictors of expression, even when they are employed to design the sequence variants as done in Cambray et al, Nat Biotech, 2018. This is discussed in L136-141.
- 2) We found some evidence that such sequence properties can improve accuracy in the low-N scenario (see Supplementary Figure S5, and comment in L145-146). The general validity of this principle is an exciting subject for future work in the field.

Further, the authors should carefully revise some of their claims on novelty and generalizability. For instance:

Q: *The fact that deep learning is data intense is widely known*

A: We agree and did not claim novelty on this. Our point is that in the synthetic biology literature, the issue of data size has not been given enough attention and is not acknowledged as a bottleneck for the application of deep learning in sequence design. Table 2 above provides concrete evidence of this point. For the avoidance of doubt, here we paraphrase two passages in our manuscript that discuss this point:

L37-42

“Although deep learning models can produce highly accurate phenotypic predictions, they come at the cost of enormous data requirements for training, typically ranging from tens to hundreds of thousands of sequences; see recent examples in Supplementary Table S1. Little attention has been paid to deep learning models in synthetic biology applications where data sizes are far below the requirements of state-of-the-art algorithms.”

L292-298

“Recent progress in high-throughput methods has led to large improvements in the size and coverage of genotype-phenotype screens, fuelling an increased interest in deep learning algorithms for phenotypic prediction. Synthetic biology offers a host of applications that would benefit from such predictors, e.g. for optimization of protein-producing strains, selection of enzymatic genes in metabolic engineering, or the design of biosensors. An often-overlooked limitation, however, is that deep learning models require huge amounts of data for training, and the sheer cost of the associated experimental work is a significant barrier for most laboratories.”

Thus, we simply argue that in synthetic biology, the use of deep learning has so far ignored the large data requirements and the high cost of acquiring such data. Ignoring such “elephant in the room” is detrimental for the adoption of deep learning in this field, because few labs have the budgets or incentives to acquire such data solely for model training.

Q: *The claim that the authors have established that “much smaller datasets than previously thought” can produce accurate predictions is simply false. Other groups (e.g. Arnold or Church amongst many others) have shown that for much fewer sequences*

A: Thank you for the comment - there is a misconception here. Our manuscript is about *prediction of protein expression from DNA sequence*, whereas the reviewer is referring to *prediction of protein function/structure from amino acid sequence*, a different and indeed well-studied problem. Prediction of protein expression has instead received comparatively little attention, let alone from a small-data perspective. Table 2 above highlights how our focus on

small data sets our work apart from the current literature. In fact the topic is so novel that there is an ongoing online challenge on the subject, where participants compete to produce the most accurate expression models from yeast promoter data.

Arnold and Church have done an enormous amount of work on machine learning for protein design and sequence-function prediction. Both groups have produced studies for small data scenarios (e.g. Biswas et al, Nat Methods 2021; Wittmann et al, Cell Systems, 2021). There are some high-level connections between that work and ours, but as said those results are for a fundamentally different task.

For clarity, in the revision we now use the term ‘sequence-to-expression models’ to distinguish our work from the literature on machine learning for protein design. We have also added citations to Wittmann et al in L241-243; Biswas et al was already cited in our first submission. We note that the Church lab has indeed produced a recent paper on prediction of protein expression (Angenent-Mari et al, Nat Commun, 2020), but that work does not have a focus on small data.

Q: *Crucially, I believe that the generalizability of some of the quantitative claims is questionable. As an example, saying that “traditional non-deep models can achieve [...] $R^2 > 70\%$ with as few as 2,000 sequences” may be correct for the underlying dataset, problem and model, but it is highly unlikely that these numbers have any general meaning (they will be different for promoters, proteins, organisms, etc).*

A: We agree that such numbers do not have a general meaning, and that it is impossible to give hard bounds on the amount of data needed because this depends on the gene construct, growth conditions and host organism. Our intention was to provide baseline performance and data sizes for the particular problem under study, and not present those as a general result applicable to a wide range of tasks.

We apologize for this confusion and have reworded the Abstract, Introduction and Discussion to better highlight our results on the relation between performance, data size and diversity, moving away from the specific metrics obtained for the particular data of Cambray et al, Nat Biotech, 2018.

Q: *Again, I believe that some of the systematic analyses are helpful and of interest for the community. However, given the points above I would recommend publication of this work in a more specialized journal after careful revision.*

A: We have responded to the reviewer’s comments and believe all concerns have been satisfactorily addressed.

General remarks:

Q: *Drawbacks of sequence design: The authors nicely point out the advantages of the way how they (or at least one coauthor) designed the sequence space in the library in their previous paper (Cambray et al).; However, designing libraries based on certain prior hypotheses also bares a risk: the introduction of user-borne bias and overfitting to the design. That should be critically discussed, especially since the authors see such effects (e.g. lines 156-161).*

A: We agree that sequence design can be problematic and bias the dataset in unforeseen ways. To mitigate such sources of bias, the original dataset in Cambray et al measured 56 independent replicates, i.e. one for each mutational series. It is hard to think that the same design artifact would arise repeatedly in different series. Moreover, we found that the biophysical properties employed for sequence design (Figure 1A) are poor predictors of expression (Figure 2B), which suggests that even if any such bias exists, it would have little impact on our analysis.

With regards to the poor generalization of the models (shown in Supplementary Figure S11), the reviewer rightly points to overfitting as a possible culprit. However, we are confident that models do not overfit and apologize for not including enough details in our first submission. We have now included details on model training, cross-validation and hyperparameter selection, all of which adhere to the best practices in machine learning. We report the cross-validation of non-deep models (Supplementary Figure S4), the training and validation loss curves of all CNNs (Supplementary Figures S6A and S7), and details on the measures we took to prevent overfitting in the Methods (e.g. use of dropout layers, early stopping, and others). For non-deep models, the cross-validation results in Supplementary Figure S4 show some variation in R^2 values for low and mid accuracy regressors, particularly those trained on the smallest datasets. But the best performing models (trained on $N=3,000$ sequences as those shown in the generalization results of Supplementary Figure S11) show only small variation and thus little evidence of overfitting. For the CNNs, the validation R^2 traces in Supplementary Figures S6A and S7 demonstrate that no overfitting is occurring.

In essence, this means that poor model generalization is not a consequence of overfitting, but rather a result of the sheer size of the sequence space (4^{96} possible sequences) and the small size of the training data. To get a sense of scale, consider that the whole Cambray et al dataset (Figure 1B) covers a tiny fraction of the entire sequence space (in the order of $10^{-51}\%$).

The first model that overcame the generalization problem appeared in an influential paper published at the time of our first submission (Vaishnav et al, Nature, 2022). That work presents a CNN of complexity similar to ours (4 convolutional layers + 2 dense layers) but trained on more than 20,000,000 promoter sequences. This suggests that improved generalization is a consequence of data size alone, not the CNN architecture. Since data of such scale are far beyond the capacity of most laboratories, we argue that poor generalization is likely to become the key limiting factor in the field. We have added new text in Discussion on these points (L357-361).

From a molecular biology perspective, the root of the problem is that we simply do not know what sequence features correlate with expression. This problem has been explored extensively in the literature by Plotkin, Cambray, Church, Claasens, Lehner, Kudla, and others. Even worse, such dependencies are context-dependent, and specific to the way the construct was designed. For example, in the case of libraries designed to perturb translation, many works have studied correlations between expression and codon usage, mRNA secondary structures, and other biophysical properties. This is because, mechanistically, we know that translational efficiency can be hampered by rare codons and that specific secondary structures (e.g. mRNA hairpins) can obstruct the ability of the ribosome to read through the transcript. Conversely, for other libraries, such as terminators (Tarnowski et al, Nat Commun, 2022), promoters (Kotopka et al, Nat Commun, 2020) and others, we do not have the sufficient mechanistic understanding to design features that are predictive of expression.

Q: *To the same end, the selection of test sets is highly critical. Since the test set follows the same modular design principles as the training data, there is a substantial risk that the models simply learn to recapitulate the design rules imposed by the user. E.g. it might detect certain sequence motifs that occur repeatedly as a result of the seed-based design approach. I believe the fact that the models are incapable to generalize across the different series could speak for this. At least the authors should test for that, e.g. by testing the models on held-out test sets that do not follow any design (randomness has its advantages there).*

A: We understand the point that full randomness can have advantages, but as discussed earlier (and in L332-337 of the manuscript), randomized libraries would lead to datasets with faraway and isolated sequences that, in turn, require a large number of variants for training accurate regressors. It is thus unclear if full randomness would provide substantial benefits, considering that these must be weighed against the cost of building, phenotyping and sequencing a large number of strains. The work in Vaishnav et al, Nature, 2022 is an excellent example of this point: they utilized a fully randomized library, but good accuracy was obtained with 20,000,000 sequences. A similar argument can be made for the 500,000 sequences employed in (Kotopka et al, Nat Commun, 2022) and other works in Table 2 above. One could even argue that in the small data scenario, careful design (i.e. not fully random) of the sequence space is in fact necessary to achieve good prediction accuracy.

While it is true that non-random libraries may cause models to learn to recapitulate the design rules imposed by the user, this is not necessarily a problem for synthetic biology tasks. Such models can still be used to discover sequences with improved properties, which is the ultimate goal of building these models in the first place. Moreover, in common synthetic biology tasks (e.g. promoter design, RBS design, etc), practitioners rarely build fully random sequences because in many cases only few positions in the sequence are mutagenized based on mechanistic knowledge. Many designs (both random and non-random) may not be functional or lead to growth defects, which further creates biases in the data and limits the benefits of fully randomized libraries.

Q: *How did the authors ensure that no overfitting is happening? I could not find corresponding analyses (e.g. the validation R²s etc.)*

A: We apologize for omitting these details in our first submission. We have now included detailed results on cross-validations of non-deep models (Supplementary Figure S4), the training and validation loss curves of all CNNs (Supplementary Figures S6A and S7), and the measures we took to prevent overfitting (Methods). We have also included more details on the CNN training in the main text (L162-193), and the new Supplementary Figure S3 shows a diagram of the construction of the training, validation and test sets, as well as of our strategy for hyperparameter optimization. We also clarify that all reported test R² values were computed on five Monte Carlo cross-validation runs, using resampled training and test sets.

Q: *The panels C and D in Figure 3 are not discussed in the main text, which leaves a gap in the story.*

A: We apologize for this oversight. We have now included a discussion on Figure 3C-D in L175-193.

Q: *The methods section is very slim, especially on ML, considering that the authors have tested “thousands of models”. I have doubts that this is presented in a way that would facilitate repetition or further use by the community in one page. A detailed ML description should be provided.*

A: We have restructured and expanded the Methods section to include more details on model training and hyperparameter optimization. Moreover, the new Supplementary Tables S2-S6 which contain the hyperparameter choices and their search spaces for all models. We have included a diagram in the new Supplementary Figure S3 explaining the construction of the training, validation, and test sets, as well as the strategies for hyperparameter optimization in both non-deep and deep models.

We went to great lengths to make the supplementary code detailed and modular, even going beyond the current practices in the synthetic biology space. In particular, we have provided jupyter notebooks implemented in Google Colab, which do not require users to install any local

libraries or dependencies. We also provided all test sets as separate csv files, so that users can reproduce the performance metrics we report in the paper; in the biological ML literature, test sets are rarely provided and instead it is left to the user to resample them, leading to variations in the results due to randomization induced by the sampling. Finally, we have had molecular biologists testing the code themselves and are confident that our results can be fully reproduced by our target audience.

Q: *Line 48: “comprehensive” seems like an overclaim*

A: We have replaced the word by ‘systematic’ as per the reviewer’s suggestion in another comment below.

Q: *Line 67: it is not clear why “random sequences obscure the effect of few specific mutations”, the meaning here should be clarified; if I understand correctly, I would argue that what the authors mean is not a result of the fact that a library is random but that it’s hard to achieve sufficient coverage for fully random libraries*

A: The reviewer is right and we have amended the text for clarity.

Q: *Figure 1C: It does not seem like a big surprise that there are five different clusters showing different distribution shapes in the phenotype (GFP measurement; side note: it’s not clear what 100% refers to, the maximum in the cluster or the entire data); How were those five selected? More importantly, it is not clear how that observation makes for an “ideally suited” dataset for ML. The authors should back the corresponding claim by a more detailed explanation and by showing the phenotypic distribution in the entire library (not five cherry-picked clusters).*

A: The five shown distributions were deliberately selected to illustrate the phenotypic diversity of the dataset. In Figure 2B we report the training results on those five series, and in Supplementary Figure S5 we show the results for all 56 mutational series. As per the reviewers’ suggestion, we have added the 56 phenotype distributions and the overall distribution in the new Supplementary Figure S2. We note that the overall distribution is approximately uniform, which highlights the advantage of having access to specific clusters with diverse phenotypic distributions as the five shown in Figure 1C.

We were surprised by the claim that these phenotypic differences are ‘no big surprise’, because there is no reason to expect that the phenotype distributions are as diverse as they are. Phenotypic diversity depends on a multitude of factors and the relation between phenotype and DNA sequence is a subject of intense study by molecular biologists. As a result, it is extremely difficult to infer the shape of the phenotype distribution from DNA sequence, let alone to expect *a priori* the level of phenotypic diversity shown in Figure 1C.

We clarify that in all distributions, 100% expression corresponds to the maximal protein expression across all variants in the dataset. We added a clarification in the caption of Figure 1C and the Methods section in L396-398.

We have also expanded on the benefits of this dataset for the purposes of comparing machine learning models in L96, and toned down the statement as suggested by the reviewer.

Q: *Figure caption of Figure 2, statement on test set: It’s not 100% clear whether there is one test set or one for each series; Also “10% from each series”, does that mean from each of the five series that were used for training, or from all 56 series? If the former is the case, there is a risk that the models perform well on the selected series but much less accurate on other series let alone on sequences that do not follow any of the design criteria in the library (see above comment on the risk of pre-designed libraries!)*

A: We apologize for the lack of clarity on this aspect; we have now included a diagram in Supplementary Figure S3A to explain how we constructed the training, validation and test sets for all models.

In Figure 2B we employed one test set for each series, containing 10% of sequences from that series. The sequences in the test set were not employed for training and in the new Supplementary Figure S3A we sketched our strategy to split the data into training, validation and test sets. As the reviewer points out, this bears the risk of models not performing well across different series, and this is indeed what we observed (Supplementary Figure S11). The focus of our paper is to explore the limits of sequence-to-expression models in small data scenarios, and therefore we deliberately designed the test sets in a way that allows us to quantify the severity of the generalization problem.

Q: *Figure 2C: the axes labels “predicted” and “measured” are not descriptive; what is the percentage (what are these values normalized to)?*

A: The fluorescence measurements were normalized to the maximum sfGFP fluorescence observed across all strains in the library. We have clarified this in the caption of Figure 1C and the Methods section (L396-398). We have also edited the axis labels in Figure 2C.

Q: *Line 136-137: “This suggest... regressors”: This phrase is meaningless without providing quantitative information, e.g. on what the authors consider as “large enough”.*

A: We have modified the statement in L178 to ‘This suggests data size alone is sufficient for training accurate regressors’.

Q: *Line 158: It’s unclear how R^2 can become negative*

A: This was explained in L229-232 of the main text, Methods A, and Eq. (1). It is a common misconception due to imprecise notation that has become unfortunately widespread. The issue has been extensively discussed in the literature; see e.g. Kvalseth T., *The American Statistician* (1985) which discusses at least eight different definitions of the R^2 statistic.

For clarity, here we restate our explanation. The R^2 score in Eq. (1) is the *coefficient of determination*, which is commonly employed as a goodness-of-fit metric in regression models. It is defined as

$$R^2 = 1 - \frac{\sum_i (y_i - f_i)^2}{\sum_i (y_i - \bar{y})^2}$$

where y_i is the i^{th} measurement, f_i is the model prediction for the i^{th} sample, and \bar{y} is the average measurement across all samples. Negative R^2 scores are thus indicative of an inadequate model structure, with worse predictions than a baseline model that simply predicts the average measurement for all samples. Note that our definition is identical to Eq. (1) in the Kvalseth paper referenced above.

Q: *198-199: the fact that deep learning requires a lot of data is by no means “often overlooked” but widely accepted and known*

A: As shown in Table 2 above, the synthetic biology literature shows a strong trend toward models trained on large datasets. While we agree that the data requirements of deep learning are well known, the synthetic literature does tend to overlook this limitation. Here we paraphrase our statement for completeness:

L292-298

Recent progress in high-throughput methods has led to large improvements in the size and coverage of genotype-phenotype screens, fuelling an increased interest in deep learning

algorithms for phenotypic prediction. Synthetic biology offers a host of applications that would benefit from such predictors, e.g. for optimization of protein-producing strains, selection of enzymatic genes in metabolic engineering, or the design of biosensors. An often-overlooked limitation, however, is that deep learning models require huge amounts of data for training, and the sheer cost of the associated experimental work is a significant barrier for most laboratories.

Q: *Lines 214-215: The authors should specify what “the literature suggest” and add corresponding references. I would argue that there is no common-sense minimal number of sequences that is required for deep learning (so the claim is false)*

A: We have added references to our claim in L317 and edited the wording to make the statement more accurate and specific to our results. We fully agree that it is impossible to establish a hard lower bound for the number of sequences required for accurate prediction, as this depends on the construct design, length, host organism and other factors.

Q: *Line 250-251: The phrase “comprehensively mapped” seems exaggerated given the sheer infinite number of combinations of dataset size, diversities and models that one could test. “systematically” would be a fair term imo.*

A: We have changed the term in L367 as per the reviewers' suggestion.

Reviewer #2

Summary:

- This study presents a comprehensive study of deep (Convolutional Neural Networks) and non-deep learning (Ridge regression, Multilayer Perceptron, Support Vector Regression, and Random Forest) models in protein expression. This work pays special attention to the data efficiency aspect of these machine learning models. The results show that deep learning models require less labeled data than usually expected from the literature.

Contributions:

- The paper does a comprehensive study of several machine learning models applied to the problem of predicting protein expression using a dataset acquired from *Escherichia coli*.

- The paper shows that deep learning models might not need as large amounts of labeled information as currently stated in the literature.

Strengths:

- This manuscript is well written and easy to follow.

- This study provides an interesting benchmark of several machine learning models for the problem of predicting protein expression.

Weaknesses:

The manuscript is interesting, but some crucial question pertaining to the main claim and contribution of the paper should be addressed before publication:

Decision:

This manuscript presents an interesting discussion about deep and non-deep learning models for protein expression. The main claim of this paper is that deep learning models do not require large amounts of data to perform well. However, more experiments should be done to confirm this claim. I suggest Major Revisions, and I encourage the authors to thoroughly review the paper to better support the main claim of the paper.

We thank the reviewer for the positive assessment of our work. Their comments were extremely insightful and prompted us to do more analyses to better support our main claims. Below we address each point in detail.

Q: *How are the hyper-parameters of all these machine learning models optimized? The right practice in machine learning is to split the data as a development and test set, where the development set is itself split into training and validation sets. The training set is used to train the model itself, for example the weights of an MLP, while the validation set is used to optimize the hyper-parameters of the model. Finally, the model is evaluated using the test set with the model trained with the best hyper-parameters from the validation set. From Section III B, we can notice that hyper-parameters "were determined via grid search and 10-fold cross validation". But, what data is used to construct the 10-folds? Is the performance of the hyper-parameter search computed using the test data? This is especially critical in deep learning models that are very prone to overfitting. If not done correctly, this issue could invalidate the main conclusion of this paper about the effectiveness of deep learning methods in low data regimes.*

A: This is a very apt comment and we apologize for the lack of clarity. All our models were trained and tested using the best practices in the field. We added the new Supplementary Figure S3 showing a schematic for the construction of the training, validation and test sets, as well as our strategy for hyperparameter optimization (grid search and 10-fold cross validation for non-deep models; Bayesian Optimization for deep models). In particular, as shown in

Supplementary Figure S3A, our validation set contains 22,400 sequences and is the aggregate of 56 smaller sets drawn from each mutational series (with 10% or ~400 sequences per series). None of the sequences in the validation set were employed for training and all performance scores were computed on test sets held out from training (also shown in the new Supplementary Figure S3). We have included more details on training and cross-validation in the Methods section, and included the cross-validation results (Supplementary Figures S4, S6A and S7).

To assuage the reviewer's concerns, we highlight that the validation R^2 traces in Supplementary Figures S6A and S7 demonstrate that our CNNs do not overfit.

Q: *What are the expected results with a deep MLP? Indeed, the final part of the CNN is an MLP of four layers. The convolutional layers have been used in the literature as a feature extraction mechanism, which has some benefits like rotation and translation invariance. It would be interesting to know how a deep MLP compares with the CNN, i.e., considering both models with a comparable number of learnable parameters.*

A: We thank the reviewer for this insightful comment. In Supplementary Figure S8 we show new training results for MLPs of increasing depth, from 3 up to 42 hidden layers. These results clearly show that the CNN outperforms the deep MLPs, even when they have a similar number of trainable parameters (both with ~2.7M trainable parameters). This strongly confirms the reviewers' intuition: the convolutional layers are acting as a feature extraction mechanism; such sequence features appear to be more predictive than the sequences themselves and thus lead to improved performance. This is a very interesting finding that nicely complements the rest of the paper and opens up new lines for future research. We have discussed these results in L194-206.

Q: *How the mixed representation is constructed in Section I B is confusing. If I understood well, the physical representation is given by an 8-dimensional vector for the whole sequence, while the one-hot encoding is a matrix of dimension $4 \times L$, where L is the dimension of the sequence. Is the biophysical vector concatenated to each amino-acid, resulting in a matrix of dimension $12 \times L$? Otherwise, what is the strategy adopted in this case?*

A: The mixed encodings were constructed from flattened one-hot encoded matrices concatenated with the vector of biophysical properties, leading to feature vectors of dimension $(4L+8)$. We have added these details to Table 1 and its caption.

Q: *Reference to different CNNs is confusing. For example in this sentence "This trend was found in all but four of the CNNs". I believe all the CNNs are the same, but trained with different amounts of data (25%, 50%, 75%). However, it is unclear what the authors refer to when discussing different CNNs. Are there more CNN architectures beside the CNN in Figure 3A? Please clarify.*

A: We indeed employed a single CNN architecture, shown in Figure 3A. Since we trained this architecture on 56 mutational series separately, and using three data sizes for each series, we regard each trained CNN as a separate model even though they all have the same architecture. Each violin plot in Figure 3C therefore shows the accuracy of 56 CNN with the same architecture but trained on different mutational series. The new paragraph in L175-193 clarifies this point.

Q: *It might not be surprising that the proposed CNN model can perform well with few labeled information. First, this CNN model is not very complex. From Figure 3A we can see that it has 3 convolutional layers, plus 4 fully connected layers. This study should report the number of learnable parameters for the CNN and the MLP. Second, we know from other fields such as Computer Vision that deep CNN with millions of parameters perform well with the ImageNet*

dataset, which contains around one million data points. I believe the ratio between data points and number of parameters in this paper should be similar to that in the ImageNet problem. There is a new emerging theory in machine learning about double descent for over-parametrized models such as CNNs. This new theory tries to explain the good performance of CNN models for low data regimes. In that sense, I believe a more interesting question is to establish how shallow or deep a CNN should be to perform well with the dataset considered in this paper. To that end, this study should perform more experiments with CNN of different complexities, maybe something like: current model, 8 convolutional layers CNN, 16 convolutional layers CNN, and 32 convolutional layers CNN. I believe this would yield a very interesting discussion.

A: We thank the reviewer for this insightful comment. The models have 58,801 (MLP) and 2,702,337 (CNN) trainable parameters; this is now said in L196, in the caption of Figure 3C and in Supplementary Figure S8A.

As suggested, in the revision we performed additional experiments with CNNs of different complexities. In Supplementary Figure S9A we show the performance of CNNs with a varying number of convolutional layers {1,2,3,5}, which clearly indicate that our choice (3 convolutional layers, Figure 3A) provides the best overall performance across mutational series. Note that instead of pushing the no. of convolutions to {8,16,32}, as suggested by the reviewer, we capped them at 5 because otherwise we are likely to make the overparameterization much worse. Instead, we opted to re-train CNNs with a variable layer width to further explore the impact of CNN complexity. These new results are shown in Supplementary Figure S9B for filter widths {32,64,128,256,512}, and clearly show that our design (width = 256) provides the best performance. These findings are within expectation, given that we designed the CNN architecture using Bayesian Optimization to maximize the validation MSE and the search space includes the number of convolutions and filter width (see Supplementary Table S4).

We thank the reviewer for the comment on the double-descent phenomenon; in a separate line of work, we are beginning to explore this on sequence-to-expression models and believe the topic will be of high interest to the machine learning community. The limited nature of biological data is a common entry barrier for machine learners to work on sequence-to-expression models and their many applications in biotechnology. The study of double-descent could open the door for the community to use more complex models and offers exciting opportunities for new methodological contributions tailored to biological data encountered in genotype-phenotype screens.

Q: *Do the classical machine learning models (Ridge regression, MLP, SVR, RF) also have a poor performance in generalization? If simpler models like RF do not have generalization issues, the problem observed with the CNN could be a problem of overfitting. Has the hyperparameters optimization been done properly (please see comment 1)?*

A: Yes, all considered models suffer from poor generalization and we apologize for not making this clear in our first submission. The new Supplementary Figure S11 contains the cross-series performance of all models considered in our paper, and clearly shows the severity of the issue. In the Methods and Supplementary Figure S3 we have clarified our strategy for hyperparameter optimization to avoid overfitting, and the results in new Supplementary Figures S4, S6A and S7 also rule out overfitting as the culprit for poor generalization. We highlight that for non-deep models (Ridge regression, MLP, SVR, RF), the cross-validation results in Supplementary Figure S4 show some variation in R^2 values for low and mid accuracy regressors, particularly those trained on the smallest datasets. But the best performing models (trained on $N=3,000$ sequences as those shown in the generalization results of Supplementary Figure S11) show only small variation and thus little evidence of overfitting.

Our conclusion is that poor generalization is an inherent challenge of sequence-to-expression models, resulting from the small size of the training data and sheer size of the sequence space (in our case, 4^{96} possible sequences). Note that we kept our training sets deliberately small because that is the focus of our paper: to explore the performance of machine learning models with datasets of realistic size. A recent paper (Vaishnav et al, Nature, 2022) showed that an off-the-shelf CNN of complexity similar to ours can indeed generalize well (4 convolutional layers + 2 dense layers), but this was achieved with a training set with more than 20,000,000 sequences. Such improved generalization performance is thus likely the result of the size of the dataset and not the model architecture itself. We have commented on this important subject in more detail in L357-361, as it means that model generalization is a key challenge for sequence-to-expression predictors in small data scenarios.

Q: *I hesitate to accept the main conclusion of this paper about CNNs requiring less labeled data than suggested in the literature. From one point of view, the Vapnik–Chervonenkis (VC) dimension in classical machine learning theory gives us some bounds in terms of the sample complexity required to solve a classification problem. We can roughly approximate the VC dimension of a neural network as the number of learnable parameters. A rule of thumb, used before the deep learning era, was to have ten times the VC dimension of data to avoid overfitting. I believe the dataset in this paper does not abide to the sample complexity given by the VC dimension for the CNN model. From another point of view, we now have the double descent theory of over-parametrized models, but having only one CNN model makes it impossible to draw some conclusions. I suggest the authors thoroughly check this affirmation, which is perhaps the most important contribution of this paper. Please also check if the other models suffer from the stated generalization issue.*

A: We thank the reviewer for yet another insightful comment. First, we clarify that our claim that ‘CNNs require fewer data than suggested in the literature’ refers to the state-of-the-art in the field of synthetic biology and sequence-to-expression modeling in general. It was not meant as a blanket statement but simply to point out that, when compared to the current literature (see new Supplementary Table 1), our study shows that CNNs do not need to be trained on expensive datasets with hundreds of thousands of DNA sequences, but they can still be useful with moderately small datasets. For example, in Figure 3C the CNN outperforms both the MLP and Random Forest models when we significantly reduce the amount of training data available.

With regards to the VC dimension, we remark that this is a measure of the capacity of a set of functions that can be learned by a binary classifier, but in our work we are focused on a regression problem. However, we agree that the question raised by R2 is an interesting one: What is the relationship between model capacity and performance and do we observe any form of "double-descent" style behavior as the model capacity increases or decreases?

In the new Supplementary Figures S8-S9 we varied the number of trainable parameters of deep MLPs and CNNs by a factor roughly comparable to the order of magnitude changes reported for ResNet and CNNs in (Nakkiran et al. ICLR 2020). While we did not find evidence of double descent behavior so far, as pointed out in the previous response, we believe this is an exciting line of work and are currently exploring the problem in much more detail.

Reviewers' Comments:

Reviewer #1:

Remarks to the Author:

I thank the authors for their response to the points raised by the other reviewer and me.

The authors have attempted to reshape their major unsubstantiated claims from the initial submission. In most cases this is attributed to a misconception by the reviewer ("we did not claim that") followed by a lengthy argumentation, which I do not share in most cases.

While individual points are of course debatable, I unfortunately must sustain my major criticism from the first revision (which holds even after the claim-reshaping): The work is insufficiently novel and does not provide any new insights, and is therefore not rightly placed in a high-impact venue such as Nat Commun.

There is no new experimental data provided and all the novelty claims do not withstand closer examination, either because they are trivial (e.g. "more diversity leads to better models", "data size is a key barrier for the adoption of deep learning in synthetic biology" etc.) or because they have been shown before ("deep learning improves performance without the need for more training data"; e.g. Hoeller et al. show that for many different trainset sizes down to 2500 etc.).

I therefore recommend publication in a more specialized (machine learning?) journal.

Reviewer #2:

Remarks to the Author:

I would like to thank the authors for the effort providing a thorough review of the manuscript. Most of my questions have been properly addressed by the authors, and I am happy to recommend this paper for acceptance. However, some minor changes should be done before publication :

The hyper-parameters of the models in Fig. S9 are optimized for the architecture with 3 layers and 256 filters, so I believe the comparison is unfair. However, I think the conclusions will be the same because of the previous hyper-parameter optimization.

It is a little bit strange that CNNs with fewer layers have more parameters than those with more layers. This seems counter-intuitive and could be confusing for non-expert machine learning readers. The authors explained that this is due to the lack of a max pooling layers. Indeed, in a CNN architecture like the one in the paper, most of the parameters are concentrated in the fully-connected (fc) layers of the architecture, so when you have few convolutional layers, and you flatten your outputs, the input to the fc layers is bigger than when you have more deep networks. That is the reason why the CNN with 1 layer has more parameters than the one with 5 layers. Please clarify this in your paper.

I am still curious about what happens with deep CNNs, for example a 16 or 32 layer CNN. I am not sure if the authors have faced computational problems, which is understandable for academic labs, or if they are facing technical challenges to train deep models. If the authors have technical challenges, I would suggest adding residual connections, since they have shown to avoid the problem of vanishing gradients. I know that CNNs are part of what we call deep learning models in the literature. However, it is a little bit strange for me to call a 3-layers CNN as a deep model. Indeed, when you look at models like residual networks with 18, 50, or even 152 layers, the deeper layers contain interesting information about the application one is solving.

Response to reviewers

“Accuracy and data efficiency in machine learning models of protein expression”

Reviewer #1

Q: I thank the authors for their response to the points raised by the other reviewer and me.

A: We thank the Referee for their comments.

Q: The authors have attempted to reshape their major unsubstantiated claims from the initial submission. In most cases this is attributed to a misconception by the reviewer ("we did not claim that") followed by a lengthy argumentation, which I do not share in most cases. While individual points are of course debatable, I unfortunately must sustain my major criticism from the first revision (which holds even after the claim-reshaping): The work is insufficiently novel and does not provide any new insights, and is therefore not rightly placed in a high-impact venue such as Nat Commun.

A: We provided a comprehensive response to each of the points raised in the first review. Without knowing which specific points are being disagreed with, it is difficult for us to understand this new criticism. In our first revision, we gave detailed replies to all concerns, including references to recent literature that highlight the novelty of our work, as well as new results and analyses (10 new figures).

It would have been helpful if citations had been provided to support the referee's concerns on the novelty and perceived triviality of our work. We gave an extensive coverage of the current literature and remain convinced that our results bring substantial novelty to the state of the art.

Q: There is no new experimental data provided and all the novelty claims do not withstand closer examination, either because they are trivial (e.g. "more diversity leads to better models", "data size is a key barrier for the adoption of deep learning in synthetic biology" etc.) or because they have been shown before ("deep learning improves performance without the need for more training data"; e.g. Hoeller et al. show that for many different training set sizes down to 2500 etc.).

I therefore recommend publication in a more specialized (machine learning?) journal.

A: We are surprised by the request for fresh experimental data at this late stage of the reviewing process. This concern was not raised by the Referee in the first round, and our first revision already incorporated the analysis of a new dataset (Figure 5 in revision, data from Vaishnav et al, Nature, 2022). We are unclear as to what specific claims need to be supported by fresh data and what specific experiments the Referee expects to see.

We already compared our work to Höllerer et al, Nat Commun, 2020, in our first rebuttal. We explained that they trained models on datasets of increasing size, from N=2,500 up to N=248,000 sequences; see Figure 4E in Höllerer et al, Nat Commun, 2020. Instead, our work explores data sizes in the opposite direction, and our *largest* data set (N=2,800) is comparable to the smallest one in Höllerer et al. **Our study deliberately focuses on data sizes much smaller than previously considered.** This was extensively discussed in page 3 of our first rebuttal and shown in Supplementary Table 1 of our manuscript. Furthermore, our focus on the interplay between accuracy, data size and sequence diversity (Figures 4-5) distinguishes our work from Höllerer et al.

Reviewer #2

Q: I would like to thank the authors for the effort providing a thorough review of the manuscript. Most of my questions have been properly addressed by the authors, and I am happy to recommend this paper for acceptance.

A: We thank the referee for careful consideration of our revised manuscript.

Q: However, some minor changes should be done before publication:

The hyper-parameters of the models in Fig. S9 are optimized for the architecture with 3 layers and 256 filters, so I believe the comparison is unfair. However, I think the conclusions will be the same because of the previous hyper-parameter optimization.

A: Yes, this is correct. Re-optimizing hyperparameters for each new CNN in Fig. S9 is beyond the scope of our work and we reiterate that, as acknowledged by the referee, our original hyperparameter optimization included the number of convolutional layers and filter width, and therefore it is expectable the deviations from the optimum lead to poorer performance.

Q: It is a little bit strange that CNNs with fewer layers have more parameters than those with more layers. This seems counter-intuitive and could be confusing for non-expert machine learning readers. The authors explained that this is due to the lack of a max pooling layers. Indeed, in a CNN architecture like the one in the paper, most of the parameters are concentrated in the fully-connected (fc) layers of the architecture, so when you have few convolutional layers, and you flatten your outputs, the input to the fc layers is bigger than when you have more deep networks. That is the reason why the CNN with 1 layer has more parameters than the one with 5 layers. Please clarify this in your paper.

A: Yes, we agree this was not explained very carefully. We have included a more comprehensive explanation in the caption of Fig. S9A following the advice of the Referee.

Q: I am still curious about what happens with deep CNNs, for example a 16 or 32 layer CNN. I am not sure if the authors have faced computational problems, which is understandable for academic labs, or if they are facing technical challenges to train deep models. If the authors have technical challenges, I would suggest adding residual connections, since they have shown to avoid the problem of vanishing gradients. I know that CNNs are part of what we call deep learning models in the literature. However, it is a little bit strange for me to call a 3-layers CNN as a deep model. Indeed, when you look at models like residual networks with 18, 50, or even 152 layers, the deeper layers contain interesting information about the application one is solving.

A: Yes, we see this point. We have now added CNNs with 8, 16 and 20 convolutional layers to Fig. S9A. We did not go further due to instabilities during the training process, but overall, the results show a deterioration of performance for deeper CNNs, in agreement with expectation given our strategy for hyperparameter optimization. In a follow up project, we are studying such deep CNNs to explore the double-descent phenomenon in much greater detail; we thank the Referee again for this insightful comment in their first refereeing report.

Reviewers' Comments:

Reviewer #2:

None